# FOUNDATION VISUAL ENCODERS ARE SECRETLY FEW-SHOT ANOMALY DETECTORS

**Guangyao Zhai**[1,2,*]    **Yue Zhou**[1,2,*]    **Xinyan Deng**[1]    **Lars Heckler-Kram**[1,3]
**Nassir Navab**[1,2]    **Benjamin Busam**[1,2]

[1]Technical University of Munich    [2]Munich Center for Machine Learning
[3]MVTec Software GmbH    [*]Equal Contribution
{firstname.lastname,b.busam}@tum.de    lars.heckler@mvtec.com

## ABSTRACT

Few-shot anomaly detection streamlines and simplifies industrial safety inspection. However, limited samples make accurate differentiation between normal and abnormal features challenging, and even more so under category-agnostic conditions. Large-scale pre-training of foundation visual encoders has advanced many fields, as the enormous quantity of data helps to learn the general distribution of normal images. We observe that the anomaly amount in an image directly correlates with the difference in the learnt embeddings and utilize this to design a few-shot anomaly detector termed FOUNDAD. This is done by learning a non-linear projection operator onto the natural image manifold. The simple operator acts as an effective tool for anomaly detection to characterize and identify out-of-distribution regions in an image. Extensive experiments show that our approach supports multi-class detection and achieves competitive performance while using substantially fewer parameters than prior methods. Backed up by evaluations with multiple foundation encoders, including fresh DINOv3, we believe this idea broadens the perspective on foundation features and advances the field of few-shot anomaly detection. Our code is at https://github.com/ymxlzgy/FoundAD.

## 1 INTRODUCTION

The variability of images is large. Understanding general concepts from pixel data is therefore by design a very complex problem. Foundation models (Zagoruyko & Komodakis, 2016; Caron et al., 2021; Zhai et al., 2023b) have provided a significant leap forward to image understanding in generalized contexts across many tasks. The design of training paradigms and the advances in computational resources enabled the computer vision community to learn powerful visual encoders whose feature embeddings are adjustable to many downstream tasks (Oquab et al., 2024). These foundation visual encoders are trained to encode samples from the core distribution of normal natural images as illustrated by the natural image manifold in Figure 1. For some vision tasks, specifically the tail end of this distribution away from the heavy core is important (Bergmann et al., 2019; Zou et al., 2022). Industrial inspection, for instance, demands robust anomaly detection techniques that can operate effectively even when limited annotated

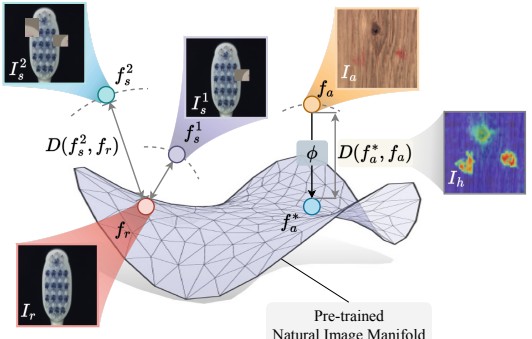

**Figure 1: Manifold Projection.** Large training sets enable foundation models to learn the manifold of natural images (illustrated schematically as a 2D surface), which lies in a higher-dimensional feature space. Normal images such as $I_r$ are embedded onto this manifold. Images with anomalies ($I_s^1$, $I_s^2$) lie further away from this manifold. The distance $D\left(f_s^i, f_r\right)$ correlates with the pixel amount of the anomaly in the image. We learn a non-linear projection operator $\phi$ that projects the embedding $f_a$ of an anomalous image $I_a$ onto its corresponding normal feature $f_a^*$. Feature comparison enables few-shot anomaly detection $I_h$.

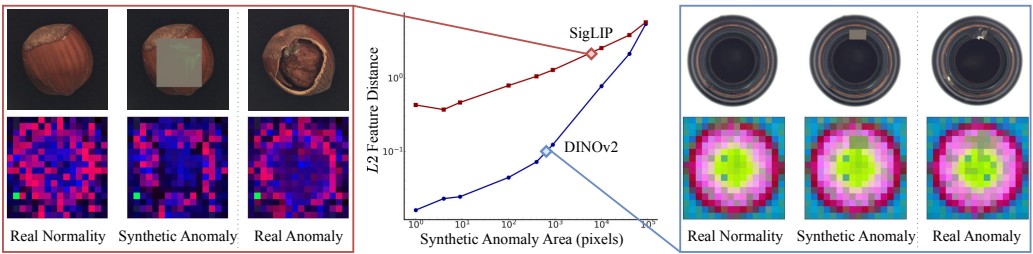

Figure 2: **Correlation of Anomaly Area with Feature Distance**. Two foundation encoders under different paradigms are shown. *Upper left/right*: Real images with corresponding synthetic and real anomalies. *Lower left/right*: Coloured PCA visualizations of their embedded features using SigLIP (Zhai et al., 2023b) (left) and DINOv2 (Oquab et al., 2024) (right). *Center*: L2-feature distance of embeddings for synthetic anomalies of increasing pixel amount on a real image. A clear correlation is visible for both foundation models.

data is available, as collecting large datasets in production is not only costly, but also impractical. This applies especially to defective samples, where the types of defects that might potentially occur are unknown prior to the launch of production. For these setups, unsupervised few-shot anomaly detection is a highly attractive approach, since it only requires a small amount of defect-free samples for training. However, the scarcity of anomalous samples generally hampers the possibility to learn comprehensive representations (Chen et al., 2020), such that traditional methods face the challenge of distinguishing subtle differences between normal and abnormal features.

Studying the embedding structure of foundation models in light of this task reveals an interesting property: The amount of anomalous area in an image directly corresponds to the feature distance in the embedding space as shown in Figure 2. This shows that foundation encoders distinguish anomaly from normality, i.e., they *"secretly detect"* the defective regions. We attribute this to the learning signal from natural images that stipulates the creation of a natural image manifold in the embedding space of a foundation model. Moving away from this structure correlates to shifting towards the tail end or outside the general distribution of normal images.

We leverage this observation in our work to design FOUNDAD, a few-shot anomaly detector that utilizes the image embeddings of a foundation visual encoder. We utilize foundational models to encode both anomalous and normal features, where structural anomalies are primitively synthesized by CutPaste (Li et al., 2021). The encoder is frozen to bootstrap the learning process by reusing the strong semantic and geometric understanding. Then, we employ a nonlinear projector to learn the necessary feature mapping with only a minimal few-shot demonstrations, such as a single sample. We validate FOUNDAD on multiple foundation visual encoders, including DINO series (Caron et al., 2021; Oquab et al., 2024; Siméoni et al., 2025), vision branches of vision-language models (VLMs) (Zhai et al., 2023b; Radford et al., 2021), and a pre-trained convolutional neural network (CNN) (Zagoruyko & Komodakis, 2016). The experiments show that FOUNDAD with DINOv3 achieves the best performance, and even it is trained on multiple classes with compact parameters, it is superior to methods that particularly focus on each class Li et al. (2024), and it performs competitively among large-scale few-shot anomaly detectors Lv et al. (2025); Zhang et al. (2025b).

Conceptually, FOUNDAD is inspired by predictive embedding approaches such as JEPA (LeCun, 2022; Assran et al., 2023) and SimSiam (Chen & He, 2021), which capture representation dependencies between paired inputs by operating purely in latent space, eschewing the need for pixel-level observation reconstruction. In contrast to these methods actively training encoders, we keep the foundation encoders frozen to enjoy the natural image manifold, and instead solely train a simple network to adapt and project the pretrained embeddings exclusively and effectively for the few-shot anomaly detection (cf. Figure 1). The entire pipeline is fast, lightweight, and easy to train. More importantly, our findings suggest that rethinking the use of foundational visual encoders can extend their applicability to complex anomaly detection tasks *without* additional design complexity, such as conventional textual prompts for assistance (Li et al., 2024; Lv et al., 2025; Zhang et al., 2025b).

In summary, the contributions of our work are threefold: **(i)** We reveal the correlation of embedding distance and anomaly amount in images for foundation visual encoders. **(ii)** Based on point (i), we introduce a feature projection method to efficiently discriminate between anomaly and normality in the embedding space. The nonlinear projector is lightweight and can be trained with minimal

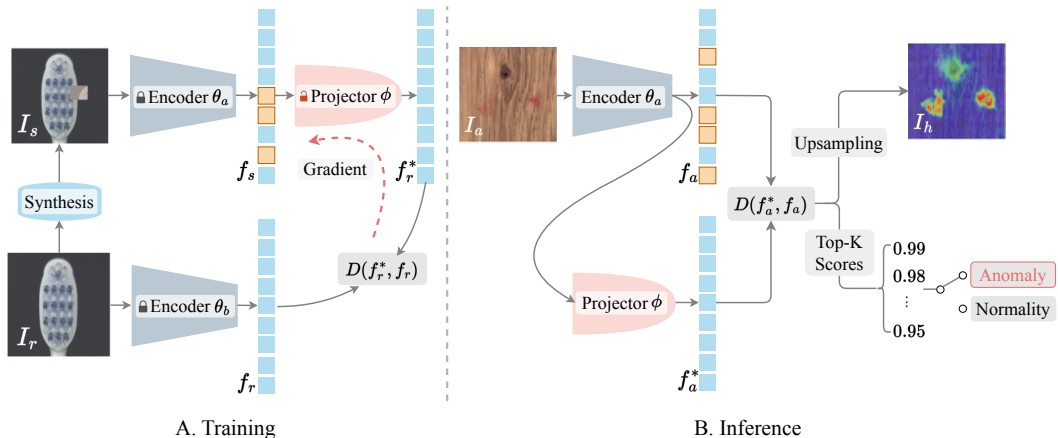

Figure 3: **A. Training pipeline.** Normal training images $I_r$ are first processed by the anomaly synthesis module to generate augmented samples $I_s$. Feature embeddings of the augmented image and the original image are extracted by the *Anomaly-Aware Encoder* $\theta_a$ and the *Reference Encoder* $\theta_b$, respectively ($\theta_a = \theta_b = \theta$). The *Manifold Projector* $\phi$ is trained to map the feature embeddings $f_s$ of the synthesized anomalous image towards the normal feature $f_r$. The training objective is to minimize the distance $D\left(f_r^*, f_r\right)$ between the projected feature $f_r^*$ and the reference feature $f_r$. **B. Inference pipeline.** During inference, an input image is processed by AE to extract feature embeddings $f_a$, which are then projected by the Projector to $f_a^*$. The anomaly score $D\left(f_a^*, f_a\right)$ for each patch is computed. We aggregate the Top-K highest patch-level anomaly scores and generate an anomaly heatmap $I_h$ by upsampling to the original image resolution.

demonstrations. **(iii)** Comprehensive experiments demonstrate that the performance of our pipeline surpasses state-of-the-art multi-class methods. More importantly, it reveals that foundation visual features *without* textual assistance suffice for few-shot anomaly detection.

## 2 METHODOLOGY

The overall architecture is illustrated in Figure 3. The network is designed to project the latent representation of either an abnormal image or a normal image to the natural image manifold learned by a foundation model. The projected representation is compared to the original representation, where large differences indicate the occurrence of anomalies. The key components of the framework include an anomaly synthesis module, two identical encoders, and a projector.

### 2.1 ANOMALY SYNTHESIS

To train the framework in an unsupervised setting, we utilize a structural anomaly synthesis module inspired by CutPaste (Li et al., 2021). As shown in Figure 2, synthesized anomalies exhibit noticeable differences from real anomalies at the pixel level, but these differences become less pronounced in the latent space. This suggests that a simple synthesis strategy is sufficient to drive anomalous features away from the natural image manifold, aligning with our projector's training objective. To enhance the reality of synthetic anomaly, we constrain anomalies to foreground regions using adaptive threshold-based binarization (Zhang et al., 2024; Yang et al., 2023).

### 2.2 VISUAL MANIFOLD PROJECTION

Our framework starts with two foundation visual encoders to extract latent representations from input images, as shown in Figure 3: the *Anomaly-Aware Encoder* (AE), which processes synthesized abnormal images $I_s$, and the *Reference Encoder* (RE), which processes original normal images $I_r$. Both encoders share the same parameters $\theta$, ensuring consistency in feature extraction and aligning embeddings within the same latent space. This design keeps normal patches in synthesized images close to their counterparts in normal images, while highlighting discrepancies in anomalous regions.

On top of these representations, we introduce the *Manifold Projector*, a nonlinear module applied after AE that maps the anomaly-aware features toward the natural image manifold with

only few-shot supervision. Since the features are tokenized, we implement the projector using a self-attention Vision Transformer (ViT). Each transformer block employs residual connections, $x_{\text{out}} = \text{Attn}(x_{\text{in}}) + x_{\text{in}}$, to stabilize training and preserve input information. Given an anomaly image $I_s$, the encoder yields an embedding $f_s = \theta(I_s)$, which deviates from the normal embedding $f_r$. The projector $\phi$ then produces a corrected embedding $f_r^*$ that aligns with the normal feature $f_r$ on the natural image manifold. Unlike reconstruction-based approaches such as (Yan et al., 2024), our method operates entirely in latent space, substantially reducing computational complexity.

## 2.3 TRAINING

The training process follows the pipeline illustrated in Figure 3. During training, given an original normal image $I_r$ in each iteration, we independently gate anomaly synthesis with a threshold $\sigma$. Let $z \sim \text{Bernoulli}(1 - \sigma)$, and then $I_s = (1 - z)\,I_r + z\,\text{Syn}(I_r)$, where Syn denotes the synthesis operation described Sec. 2.1. We obtain feature embeddings using $\theta$: $f_s = \theta(I_s), f_r = \theta(I_r)$. Then the projector $\phi$ maps the synthesized anomaly feature $f_s$ to the estimated normal feature: $f_r^* = \phi(f_s)$. The training objective is to minimize the discrepancy between the projected feature and the reference normal feature via:

$$\mathcal{L} = D(f_r^*, f_r) = \frac{1}{N} \sum_{i=1}^{N} (f_{r,i}^* - f_{r,i})^2, \tag{1}$$

where $\mathcal{L}$ is the $L2$ loss, and $N$ is the number of embedded patches.

## 2.4 INFERENCE

Given a test image $I_a$, its feature embedding is extracted using AE as $f_a = \theta(I_a)$. The projector $\phi$ is then applied to map the extracted feature embedding to the estimated feature on the natural image manifold $f_a^* = \phi(f_a)$. We compute the patch-level anomaly score as the squared $L2$ distance between the extracted feature and its projected counterpart, turning Equation 1 to:

$$S_{\text{patch}} = D(f_a^*, f_a) = \frac{1}{N} \sum_{i=1}^{N} (f_{a,i}^* - f_{a,i})^2. \tag{2}$$

For image-level anomaly detection, we compute the average of the Top-K patch anomaly scores:

$$S_{\text{image}} = \frac{1}{K} \sum_{i=1}^{K} S_{\text{patch},i}. \tag{3}$$

Finally, to generate pixel-level anomaly maps, the patch-level scores are upsampled to the original image resolution, highlighting regions with the most significant deviations.

# 3 EXPERIMENTS

## 3.1 EXPERIMENTAL SETUP

**Datasets** Following the protocol in IIPAD (Lv et al., 2025), we evaluate FOUNDAD with various foundation encoders on two popular industrial anomaly detection datasets: MVTec-AD (Bergmann et al., 2019) and VisA (Zou et al., 2022). MVTec-AD consists of $5,354$ high-resolution images across five texture and ten object categories, with $1,725$ for testing, covering various real-world defects such as contamination and structural deformations. VisA includes twelve object categories with $10,821$ images, featuring $9,621$ normal samples and $1,200$ anomalous images. It presents additional challenges due to complex structures, multi-instance objects, and diverse anomaly patterns, requiring models to generalize across varying levels of complexity. We provide additional results on two more datasets, BTAD (Mishra et al., 2021) and DTD (Cimpoi et al., 2014), where BTAD focuses on body and surface defects of industrial products, and DTD specializes in textures.

**Evaluation Metrics** To comprehensively assess anomaly detection and localization performance, we employ three standard metrics: (1) Area Under the Receiver Operating Characteristic Curve

(AUROC), which evaluates the model's ability to distinguish normal and anomalous samples at both image and pixel levels; (2) Area Under the Precision-Recall Curve (AUPR), which is particularly effective for imbalanced datasets by emphasizing precision-recall trade-offs; and (3)Per-Region-Overlap (PRO), which measures region-level anomaly localization by computing the overlap between predicted and ground-truth anomalous regions. Curve areas involving the false positive rate are calculated only up to an FPR of 0.3 (Bergmann et al., 2019). These metrics collectively ensure a robust evaluation of both detection accuracy and localization effectiveness.

**Baseline**  We evaluate the method against state-of-the-art anomaly detection methods across multi/one-class settings. For few-shot and multi-class scenarios, we compare with FastRecon (Fang et al., 2023), AnomalySD (Yan et al., 2024), and IIPAD (Lv et al., 2025). Additionally, we include one-class methods such as WinCLIP (Jeong et al., 2023), InCTRL (Zhu & Pang, 2024), Anomaly-CLIP (Zhou et al., 2024), PromptAD (Li et al., 2024), LogSAD (Zhang et al., 2025b), and Adapt-CLIP (Gao et al., 2025), as well as classical models like SPADE (Cohen & Hoshen, 2020) and PatchCore (Roth et al., 2022), referring to results reported in IIPAD.

**Implementation Details**  FOUNDAD is compatible with various encoders, and we report the one backed by a pretrained DINOv3 ViT-B (Siméoni et al., 2025) here. For details about more encoders, please refer to the Supplementary Material A. The input images are resized to $512 \times 512 \times 3$ for consistency across experiments. The projector is implemented as a ViT with a depth of 6 layers. For stable and efficient optimization, we utilize the Adam optimizer with a weight decay of $1 \times 10^{-4}$. The learning rate is set to $0.001$, empirically chosen to ensure convergence. All experiments are conducted on a single RTX 3090 GPU with a batch size of $8$. During training, we set the synthesis threshold $\sigma$ as $0.5$. For inference, the image-level anomaly score is computed based on the Top-K highest patch anomaly scores. $K$ is set to 10 for MVTec-AD, BTAD, and DTD, and 6 for VisA.

Table 1: Comparison of results on MVTec-AD and VisA against various **multi-class-one-model** few-shot methods. Metrics include image-level AUROC (%), AUPR (%), and pixel-level AUROC (%), PRO (%). We color the best and the second in the k-shot setting.

| Shot | Method | w/o Texts | MVTec-AD | | | | VisA | | | |
|------|--------|-----------|----------|----------|----------|------|----------|----------|----------|------|
| | | | I-AUROC | AUPR | P-AUROC | PRO | I-AUROC | AUPR | P-AUROC | PRO |
| | SPADE | ✓ | 58.8 | 63.7 | 60.4 | 53.1 | 61.3 | 68.2 | 69.0 | 57.2 |
| | PatchCore | ✓ | 63.7 | 81.2 | 83.9 | 72.7 | 58.9 | 62.8 | 76.7 | 64.3 |
| | FastRecon | ✓ | 51.2 | 72.6 | 62.1 | 60.3 | 55.0 | 72.8 | 70.7 | 58.2 |
| | WinCLIP | ✗ | 92.8 | 96.5 | 92.4 | 83.5 | 83.1 | 85.1 | 94.6 | 80.9 |
| 1 | PromptAD | ✗ | 86.3 | 93.4 | 91.8 | 83.6 | 80.8 | 83.2 | 96.3 | 82.2 |
| | AnomalySD | ✗ | 93.6 | 96.9 | 94.8 | 89.2 | 86.1 | 89.1 | 96.5 | 93.9 |
| | IIPAD | ✗ | 94.2 | 97.2 | 96.4 | 89.8 | 85.4 | 87.5 | 96.9 | 87.3 |
| | FOUNDAD (**Ours**) | ✓ | 96.1 | 97.9 | 96.8 | 92.8 | 92.6 | 92.0 | 99.7 | 98.0 |
| | SPADE | ✓ | 68.4 | 84.2 | 61.2 | 54.7 | 66.8 | 72.0 | 71.3 | 59.6 |
| | PatchCore | ✓ | 72.4 | 86.2 | 89.6 | 74.2 | 60.2 | 64.3 | 82.4 | 68.1 |
| | FastRecon | ✓ | 51.7 | 74.9 | 62.4 | 59.9 | 58.2 | 74.6 | 79.6 | 63.5 |
| | WinCLIP | ✗ | 92.7 | 96.3 | 92.4 | 83.9 | 83.7 | 84.9 | 95.1 | 81.8 |
| 2 | PromptAD | ✗ | 89.2 | 94.8 | 92.2 | 84.3 | 84.3 | 87.8 | 96.9 | 84.7 |
| | AnomalySD | ✗ | 94.8 | 97.0 | 95.8 | 90.4 | 87.4 | 90.1 | 96.8 | 94.1 |
| | IIPAD | ✗ | 95.7 | 97.9 | 96.7 | 90.3 | 86.7 | 88.6 | 97.2 | 87.9 |
| | FOUNDAD (**Ours**) | ✓ | 96.8 | 98.3 | 97.0 | 93.3 | 93.5 | 93.0 | 99.7 | 98.0 |
| | SPADE | ✓ | 76.6 | 88.8 | 62.8 | 55.6 | 73.0 | 76.6 | 72.1 | 60.9 |
| | PatchCore | ✓ | 74.9 | 88.8 | 92.6 | 80.8 | 62.6 | 69.9 | 85.4 | 70.6 |
| | FastRecon | ✓ | 50.8 | 73.1 | 65.0 | 62.8 | 57.6 | 73.7 | 78.8 | 62.9 |
| | WinCLIP | ✗ | 94.0 | 96.9 | 92.9 | 84.4 | 84.1 | 86.1 | 95.2 | 82.1 |
| 4 | PromptAD | ✗ | 90.6 | 96.5 | 92.4 | 84.6 | 85.7 | 88.8 | 97.2 | 84.7 |
| | AnomalySD | ✗ | 95.6 | 97.6 | 96.2 | 90.8 | 88.9 | 90.9 | 97.5 | 94.3 |
| | IIPAD | ✗ | 96.1 | 98.1 | 97.0 | 91.2 | 88.3 | 89.6 | 97.4 | 88.3 |
| | FOUNDAD (**Ours**) | ✓ | 97.1 | 98.6 | 97.2 | 93.5 | 94.4 | 94.0 | 99.7 | 98.4 |

The multi-class and few-shot results of SPADE (Cohen & Hoshen, 2020), PatchCore (Roth et al., 2022), FastRecon (Fang et al., 2023), WinCLIP, and PromptAD are from IIPAD (Lv et al., 2025).

## 3.2 EXPERIMENTAL RESULTS

For each few-shot setting, we conduct three different sample combinations, each drawn under a random seed. The averaged results on the MVTec-AD and VisA across different few-shot settings are reported in Table 1. Some one-class-one-model full-shot methods (e.g., SPADE, PatchCore) from IIPAD are adapted to the multi-class-one-model few-shot setting for fair comparison. To further

demonstrate the competitiveness of FOUNDAD, Table 2 and Table 3 present results on all datasets under the one-class-one-model paradigm. Detailed outcomes from each run are provided in Supplementary Material C.1.

**Quantitative Comparison with Multi-Class Baselines** As shown in Table 1, FOUNDAD consistently achieves the best performance across both image-level classification and pixel-level segmentation metrics. Classical few-shot methods such as SPADE, PatchCore, and FastRecon suffer from severe performance degradation when extended to the multi-class setting, highlighting the difficulty of adapting them to various categories. In contrast, FOUNDAD handles multi-class adaptation effectively with its simple projector architecture, even under limited training data. Notably, FOUNDAD achieves strong results *without relying on text prompts*, distinguishing it from recent prompt-based baselines. On MVTec-AD, FOUNDAD shows a clear advantage under the low-shot regime. For instance, in the 1-shot case, it surpasses the second-best method IIPAD by $1.9\%$ in I-AUROC and $3.0\%$ in PRO. On VisA, FOUNDAD delivers nearly perfect localization, with pixel-level AUROC reaching $99.7\%$ consistently across all few-shot settings, outperforming IIPAD by up to $2.8\%$. As the number of shots increases to 2 and 4, FOUNDAD not only maintains competitive image-level accuracy but also further strengthens its lead in localization metrics. These results highlight the robustness and strong generalization ability of FOUNDAD in the challenging multi-class few-shot anomaly detection scenario.

Table 2: Comparison of results on MVTec-AD and VisA against various **one-class-one-model** few-shot methods. Ours remains the **multi-class-one-model** few-shot setting. Metrics include image-level AUROC (%), AUPR (%), pixel-level P-AUROC (%), and PRO (%). We color the best and the second in the k-shot setting.

| Shot | Method | w/o Texts | MVTec-AD | | | | VisA | | | |
|---|---|---|---|---|---|---|---|---|---|---|
| | | | I-AUROC | AUPR | P-AUROC | PRO | I-AUROC | AUPR | P-AUROC | PRO |
| 1 | SPADE | ✓ | 81.0 | 90.6 | 91.2 | 83.9 | 79.5 | 82.0 | 95.6 | 84.1 |
| | PatchCore | ✓ | 83.4 | 92.2 | 92.0 | 79.7 | 79.9 | 82.8 | 95.4 | 80.5 |
| | WinCLIP | ✗ | 93.1 | 96.5 | 95.2 | 87.1 | 83.8 | 85.1 | 96.4 | 85.1 |
| | InCTRL[1] | ✗ | 91.3 | 95.2 | 94.6 | 87.8 | 83.2 | 84.1 | 89.0 | 66.7 |
| | AnomalyCLIP[2] | ✗ | 95.2 | 97.2 | 94.6 | 87.6 | 87.7 | 87.7 | 83.2 | 90.1 |
| | PromptAD | ✗ | 94.6 | 97.1 | 95.9 | 87.9 | 86.9 | 88.4 | 96.7 | 85.1 |
| | LogSAD[3] | ✗ | 95.5 | 97.3 | 97.0 | 92.5 | 89.8 | 90.3 | 97.5 | 88.2 |
| | AdaptCLIP | ✗ | 94.5 | 97.5 | 94.3 | 89.5 | 90.5 | 92.3 | 96.8 | 90.7 |
| | FOUNDAD (**Ours**) | ✓ | 96.1 | 97.9 | 96.8 | 92.8 | 92.6 | 92.0 | 99.7 | 98.0 |
| 2 | SPADE | ✓ | 82.9 | 91.7 | 92.0 | 85.7 | 80.7 | 82.3 | 96.2 | 85.7 |
| | PatchCore | ✓ | 86.3 | 93.8 | 93.3 | 82.3 | 81.6 | 84.8 | 96.1 | 82.6 |
| | WinCLIP | ✗ | 94.4 | 97.0 | 96.0 | 88.4 | 84.6 | 85.8 | 96.8 | 86.2 |
| | InCTRL[1] | ✗ | 91.8 | 95.5 | 95.2 | 88.3 | 86.3 | 86.8 | 89.8 | 68.1 |
| | AnomalyCLIP[2] | ✗ | 95.4 | 97.3 | 94.9 | 87.8 | 87.8 | 89.1 | 84.5 | 90.8 |
| | PromptAD | ✗ | 95.7 | 97.9 | 96.2 | 88.5 | 88.3 | 90.0 | 97.1 | 85.8 |
| | LogSAD[3] | ✗ | 96.3 | 97.6 | 97.3 | 93.1 | 91.8 | 92.0 | 97.8 | 89.7 |
| | AdaptCLIP | ✗ | 95.7 | 97.9 | 94.5 | 90.3 | 92.2 | 93.6 | 97.1 | 91.2 |
| | FOUNDAD (**Ours**) | ✓ | 96.9 | 98.3 | 97.0 | 93.2 | 93.8 | 93.3 | 99.7 | 98.2 |
| 4 | SPADE | ✓ | 84.8 | 92.5 | 92.7 | 87.0 | 81.7 | 83.4 | 96.6 | 87.3 |
| | PatchCore | ✓ | 88.8 | 94.5 | 94.3 | 84.3 | 85.3 | 87.5 | 96.8 | 84.9 |
| | WinCLIP | ✗ | 95.2 | 97.3 | 96.2 | 89.0 | 87.3 | 88.8 | 97.2 | 87.6 |
| | InCTRL[1] | ✗ | 93.1 | 96.3 | 95.8 | 89.5 | 87.8 | 88.0 | 90.2 | 68.8 |
| | AnomalyCLIP[2] | ✗ | 96.1 | 97.8 | 95.5 | 88.2 | 88.8 | 90.1 | 85.2 | 91.4 |
| | PromptAD | ✗ | 96.6 | 98.5 | 96.5 | 90.5 | 89.1 | 90.8 | 97.4 | 86.2 |
| | LogSAD[3] | ✗ | 96.6 | 97.7 | 97.5 | 93.5 | 93.2 | 93.4 | 98.1 | 90.5 |
| | AdaptCLIP | ✗ | 96.6 | 98.4 | 94.8 | 90.9 | 93.1 | 94.3 | 97.3 | 91.7 |
| | FOUNDAD (**Ours**) | ✓ | 97.1 | 98.6 | 97.2 | 93.6 | 94.4 | 94.0 | 99.7 | 98.4 |

[1] InCTRL (Zhu & Pang, 2024) only targeted at the image-level anomaly detection, yet we report pixel-level metrics for readers' comprehension. [2] AnomalyCLIP (Zhou et al., 2024) was originally designed for zero-shot detection. Following AdaptCLIP (Gao et al., 2025), we extend it to support detection with few-shot references. [3] LogSAD originally only provides I-/P-AUROC, and we reproduce the experiment in three rounds and report averaged results on every metric, adhering to our protocols.

**Quantitative Comparison with One-Class Baselines** As shown in Table 2, even against few-shot baselines specialized for the one-class-one-model setting, FOUNDAD consistently achieves top performance on both MVTec-AD and VisA. In the 1-shot setting, our method already surpasses strong baselines PromptAD and the previous state-of-the-art LogSAD, with a particularly large margin on VisA by $2.2\%$ in P-AUROC and $9.8\%$ in PRO over LogSAD. In the 4-shot scenario on MVTec-AD, FOUNDAD outperforms PromptAD by $0.7\%$ in P-AUROC and $3.1\%$ in PRO, while achieving slightly lower P-AUROC than LogSAD. On VisA, however, FOUNDAD surpasses LogSAD by $1.6\%$

in P-AUROC and **7.9**% in PRO, demonstrating consistent superiority in both classification and localization. Moreover, FOUNDAD relies on a single visual foundation model, which makes it more efficient than LogSAD's three-model DINO-CLIP-SAM pipeline. Importantly, our improvements are achieved in the more challenging *multi-class-one-model* setting, whereas other baselines operate under the simpler one-class formulation with class-specific memory banks or text prompts. This underscores the robustness and learning ability of FOUNDAD, which remains effective even under harder problem constraints. As shown in Table 3, FOUNDAD continues to achieve state-of-the-art results on the BTAD and DTD datasets, consistently securing top performance across all metrics against various strong few-shot baselines. For example, in the 1-shot setting, our method significantly outperforms the closest competitor, AdaptCLIP, on BTAD, yielding a notable **3.0**% improvement in I-AUROC and a substantial **6.8**% boost in PRO. On the DTD dataset, FOUNDAD similarly exceeds the strong localization baseline InCTRL by 1.9% in PRO.

Table 3: Comparison of results on BTAD and DTD against various **one-class-one-model** few-shot methods. Ours remains the **multi-class-one-model** few-shot setting. Metrics include image-level AUROC (%), AUPR (%), pixel-level P-AUROC (%), and PRO (%). We color the best and the second in the k-shot setting.

| Shot | Method | w/o Texts | BTAD | | | | DTD | | | |
|---|---|---|---|---|---|---|---|---|---|---|
| | | | I-AUROC | AUPR | P-AUROC | PRO | I-AUROC | AUPR | P-AUROC | PRO |
| 1 | WinCLIP | ✗ | 84.4 | 80.5 | 95.6 | 66.6 | 97.9 | 99.0 | 96.5 | 89.7 |
| | InCTRL | ✗ | 88.5 | 83.4 | 96.6 | 71.0 | 97.9 | 98.9 | 98.6 | 95.6 |
| | AnomalyCLIP | ✗ | 88.5 | 74.2 | 95.3 | 75.5 | 98.0 | 99.2 | 97.6 | 93.5 |
| | AdaptCLIP | ✗ | 93.4 | 95.8 | 96.6 | 77.3 | 98.0 | 99.1 | 97.4 | 92.8 |
| | FOUNDAD (**Ours**) | ✓ | 96.4 | 96.5 | 98.1 | 84.1 | 99.2 | 99.7 | 98.9 | 97.5 |
| 2 | WinCLIP | ✗ | 85.8 | 82.5 | 95.7 | 66.8 | 98.1 | 99.1 | 96.6 | 89.8 |
| | InCTRL | ✗ | 88.5 | 83.4 | 96.6 | 71.0 | 98.3 | 99.1 | 98.7 | 95.6 |
| | AnomalyCLIP | ✗ | 89.2 | 75.4 | 95.5 | 76.0 | 98.2 | 99.3 | 97.7 | 93.7 |
| | AdaptCLIP | ✗ | 93.4 | 95.9 | 96.7 | 77.8 | 97.4 | 99.2 | 97.6 | 92.9 |
| | FOUNDAD (**Ours**) | ✓ | 97.3 | 96.8 | 98.2 | 83.3 | 99.2 | 99.7 | 98.9 | 97.5 |
| 4 | WinCLIP | ✗ | 87.8 | 88.1 | 95.9 | 67.0 | 98.2 | 99.2 | 96.8 | 90.4 |
| | InCTRL | ✗ | 67.5 | 80.9 | 96.8 | 71.4 | 97.7 | 98.3 | 98.7 | 95.9 |
| | AnomalyCLIP | ✗ | 90.5 | 77.5 | 95.7 | 76.6 | 98.4 | 99.4 | 97.8 | 93.8 |
| | AdaptCLIP | ✗ | 93.3 | 96.4 | 96.8 | 78.2 | 98.5 | 99.3 | 97.8 | 93.2 |
| | FOUNDAD (**Ours**) | ✓ | 95.5 | 96.7 | 98.3 | 82.7 | 99.3 | 99.7 | 99.0 | 97.6 |

**Qualitative Comparison**    We present qualitative comparisons in Figure 4. FOUNDAD effectively localizes anomalous regions with high precision on both datasets. It precisely localizes anomalous regions, successfully capturing both large structural defects and subtle, fine-grained anomalies. In contrast to other methods, FOUNDAD yields cleaner segmentations with substantially less noise in background areas. Additional examples are provided in Supplementary Material C.2 and C.5.

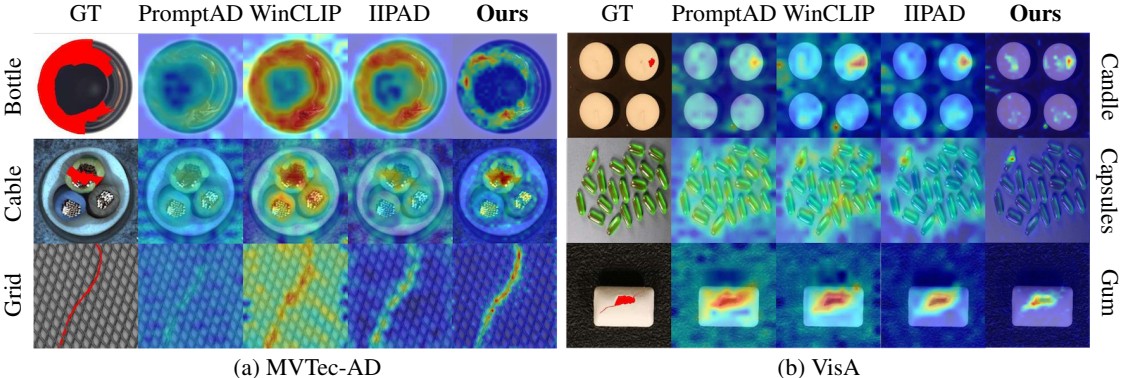

(a) MVTec-AD                                          (b) VisA

Figure 4: **Qualitative comparison with few-shot baselines in 1-shot setting.** We directly compare our results with the ones cropped from IIPAD (Lv et al., 2025).

**Inference Efficiency** We report the inference time and memory consumption following (Batzner et al., 2024). Our projector consists of 11.8 M trainable parameters. With DINOv3 as the backbone, FOUNDAD contains 97.8 M parameters in total and achieves an average inference time of 128.7 ms per image, corresponding to a throughput of approximately 7.8 images per second, with a peak memory consumption of 1, 386 MiB on a single RTX 3090. To assess the inference efficiency, we analyze the trade-off between performance and model size across various representative baselines. As shown in Figure 5, FOUNDAD achieves overall the best accuracy while using at least one order of magnitude fewer parameters than the large models LogSAD ($\approx$ **13.3**$\times$) and IIPAD ($\approx$ **10.3**$\times$). Despite the compact size, FOUNDAD maintains desirable efficiency, especially suitable in industrial usages.

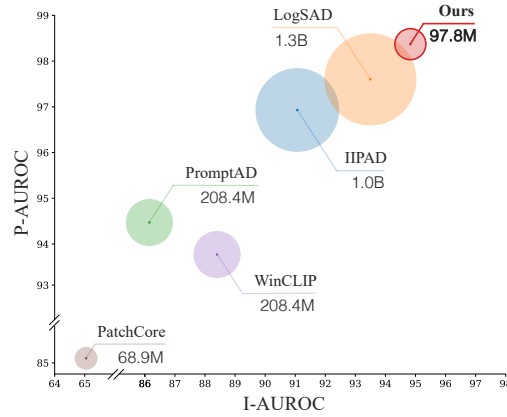

Figure 5: Bubble chart of AUROC results across different methods, averaged over MVTec-AD and VisA from Table 1. The smaller the circle is, the fewer parameters it has.

## 3.3 ABLATION STUDY

**Comparison between DINOv3 Layers** We analyze the effect of selecting different layers in DINOv3 for feature extraction in the 1-shot setting on MVTec-AD, as shown in Figure 6. Layer 10, used in FOUNDAD, achieves the overall best performance, with the highest I-AUROC (96.1%), AUPR (97.9%), and P-AUROC (96.8%), and a competitive PRO (92.8%). While adjacent layers such as layer 9 and 11 yield comparable results, performance declines when using shallower (e.g., layer 6) or deeper layers (e.g., layer 12), especially in terms of I-AUROC and PRO. This indicates that mid-to-late layers strike a better balance between semantics and spatial precision, while overly abstract or low-level features are suboptimal for fine-grained localization.

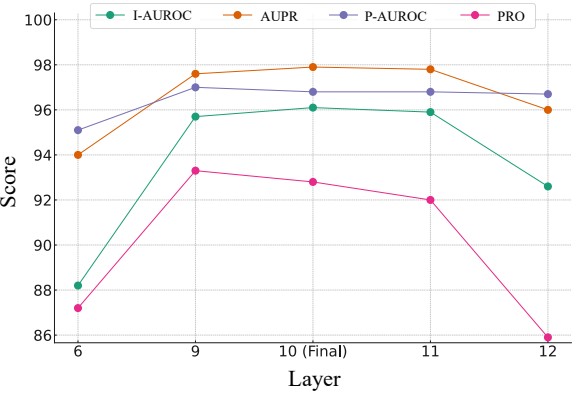

Figure 6: Comparison of performance using different layers of DINOv3 in a 1-shot setting on MVTec-AD.

Table 4: Comparison of performance with different encoders as the backbones in a 1-shot setting on MVTec-AD. We color the best and the second .

|  | DINOv3 | DINOv2 | DINOSigLIP | DINO | SigLIP | CLIP | WideResnet |
|---|---|---|---|---|---|---|---|
| Pre-trained w/o Texts | ✓ | ✓ | ✗ | ✓ | ✗ | ✗ | ✓ |
| I-AUROC | 96.1 | 95.2 | 92.5 | 88.3 | 87.8 | 79.0 | 73.1 |
| AUPR | 97.9 | 97.4 | 95.1 | 94.2 | 93.8 | 87.9 | 87.2 |
| P-AUROC | 96.8 | 96.4 | 93.1 | 96.2 | 86.0 | 90.9 | 89.4 |
| PRO | 92.8 | 92.5 | 87.2 | 87.8 | 71.1 | 70.9 | 75.6 |

**Comparison of Different Foundation Models** To further investigate the capability of different visual encoders on FOUNDAD, we conduct an ablation study comparing several commonly used encoders in the 1-shot setting on MVTec-AD, as shown in Table 4. For VLMs, we utilize only the vision backbone to extract features in the latent space. Our results demonstrate that DINOv3 achieves the best overall performance. Among the alternative encoders, the DINO series remains competitive, with DINOv2 and DINOSigLIP reaching I-AUROC scores of 95.2% and 92.5%, respectively. As expected, CLIP lags markedly, achieving only 70.9% PRO, consistent with WinCLIP's observation that CLIP lacks pixel-level information (Jeong et al., 2023). This also suggests that CLIP is less

effective than SigLIP in learning natural image manifold for fine-grained anomaly localization in few-shot settings. WideResNet shows the weakest performance, indicating that traditional CNNs, even pretrained, struggle to generalize well under minimal supervision.

Most importantly, Table 4 reveals that foundation features pre-trained from pure visual supervision, without alignment on textual information, can still deliver highly competitive performance for anomaly detection in challenging few-shot settings. This finding highlights that textual features are not a necessity; instead, the representational power of strong visual features alone can be fully leveraged to uncover anomalies with minimal data.

**Comparison of Projector Designs**  In our initial exploration of projector architectures, we compared two widely used modules, MLP layers and ViT attention blocks, under different layer configurations in the 1-shot setting on MVTec-AD. As shown in Table 5, ViT consistently outperforms MLP with the same depth. This advantage arises from the self-attention mechanism, which enables richer patchwise interactions and improves the detection of fine-grained anomalies. However, simply increasing the network depth does not necessarily yield further gains and instead adds computational overhead. Guided by these observations, we adopt a 6-layer self-attention block in our final design, as it offers a favorable trade-off between accuracy and computational cost.

Table 5: 1-shot performance comparison of different architectures of the manifold projector on MVTec-AD.

| Type | Depth | I-AUROC | AUPR | P-AUROC | PRO |
|------|-------|---------|------|---------|-----|
| ViT  | 4 | 95.5 | 97.2 | 96.6 | 92.6 |
|      | 6 | 96.1 | 97.9 | 96.8 | 92.8 |
|      | 8 | 95.8 | 97.3 | 96.8 | 92.5 |
| MLP  | 4 | 93.5 | 96.2 | 95.7 | 91.2 |
|      | 6 | 92.1 | 95.4 | 95.2 | 90.7 |
|      | 8 | 87.8 | 91.9 | 91.2 | 82.1 |

**Top-K Selection**  The optimal $K$ varies across datasets, as the size and distribution of abnormal regions differ among various datasets (Sträter et al., 2024). We evaluate the effect of different values of $K$ in the Top-K selection mechanism on MVTec-AD and VisA in a 1-shot setting. Table 6 indicates that increasing $K$ initially improves performance, with the best results observed at $K = 10$ for MVTec-AD and $K = 6$ for VisA. Beyond these points, performance gradually declines.

Table 6: Comparison of performance with DINOv3 backbone and different $K$ of Top-$K$ in a 1-shot setting on MVTec-AD and VisA. We color the  best .

| Dataset | Metric | K | | | | | | | |
|---------|--------|-------|-------|-------|-------|-------|-------|-------|-------|
|         |        | 1 | 2 | 4 | 6 | 10 | 14 | 16 | 20 |
| MVTec-AD | I-AUROC | 94.82 | 95.41 | 95.89 | 95.96 | 96.09 | 95.91 | 95.86 | 95.74 |
|          | AUPR    | 97.10 | 97.47 | 97.78 | 97.80 | 97.92 | 97.81 | 97.78 | 97.75 |
| VisA    | I-AUROC | 91.64 | 92.26 | 92.63 | 92.64 | 92.44 | 92.28 | 92.17 | 91.96 |
|         | AUPR    | 90.91 | 91.39 | 91.56 | 91.98 | 91.56 | 91.55 | 91.53 | 91.41 |

## 4 RELATED WORK

**Multi-class Anomaly Detection**  Many existing methods follow a one-class-one-model (Yi & Yoon, 2020; Cohen & Hoshen, 2020; Defard et al., 2021; Roth et al., 2022; Rudolph et al., 2021; Zavrtanik et al., 2021; 2022; Deng & Li, 2022; Tien et al., 2023; Iqbal et al., 2024; Zhang et al., 2024; Li et al., 2021; Hu et al., 2024; Zhang et al., 2023) paradigm, necessitating a separate model for each class, which limits generalization to unseen categories. To overcome this limitation, general models (Huang et al., 2022; 2024; He et al., 2024a) and multi-class-one-model approaches (You et al., 2022; Zhao, 2023; Lu et al., 2023; He et al., 2024b) have been proposed. In this work, we focus primarily on multi-class settings. UniAD (You et al., 2022) presents a unified framework capable of handling multiple classes simultaneously, while HVQ-Trans (Lu et al., 2023) employs vector quantization to learn a structured latent space that enhances anomaly detection within each category and preserves cross-category discrimination. DiAD (Zhang et al., 2025a) introduces a diffusion-based method for multi-class anomaly detection. Notably, these methods typically require a large amount of training

data, which is impractical in real industrial scenarios since collecting and labeling samples is challenging and expensive. Therefore, we aim to develop an approach for multi-class anomaly detection with minimal training effort.

**Few-shot Anomaly Detection** Few-shot anomaly detection aims to detect and localize anomalies using minimal training samples per category (Fang et al., 2023; Duan et al., 2023; Damm et al., 2024; Zhu & Pang, 2024; Huang et al., 2022; 2024; Jeong et al., 2023; Gu et al., 2024b; Yan et al., 2024; Lv et al., 2025). For example, RegAD (Huang et al., 2022) introduces spatial transformation consistency to ensure robustness under few-shot conditions, while FastRecon (Fang et al., 2023) leverages an efficient feature reconstruction approach for rapid adaptation to unseen anomaly types. Another line of research exploits textual information to support few-shot anomaly detection; methods such as AnomalyGPT (Gu et al., 2024b), InCTRL (Zhu & Pang, 2024), WinCLIP (Jeong et al., 2023), and PromptAD (Li et al., 2024) employ vision-language models to address the few-shot detection problem. Moreover, recent work on the multi-class setting, such as AnomalySD (Yan et al., 2024) and IIPAD (Lv et al., 2025), has proposed unified models that handle multiple categories within a single framework while achieving competitive performance. LogSAD (Zhang et al., 2025b) leverages GPT-4V (Achiam et al., 2023) to generate textual reasoning rules per category and combines patch-level and set-level detections with score calibration, achieving strong performance in a training-free manner. However, relying on textual inputs or support images during inference remains impractical in many real-world applications.

**Foundation Models for Visual Understanding** Foundation models have driven major advances in visual understanding (Zhou et al., 2022; Gu et al., 2024a; Liu et al., 2024), content generation (Zhai et al., 2024; 2023a; Karras et al., 2023), vision-based planning (Zhou et al., 2025; Kim et al., 2025), geometry estimation (Chen et al., 2024; Wang et al., 2025; Edstedt et al., 2024), and future prediction (Karypidis et al., 2025; Baldassarre et al., 2025). Early supervised pretraining on large-scale datasets such as ImageNet (Deng et al., 2009) with convolutional networks (He et al., 2016; Zagoruyko & Komodakis, 2016) established strong transferable features, while self-supervised contrastive learning, such as SimCLR (Chen et al., 2020), MoCo (He et al., 2020), further improved label-free representation learning. Vision Transformers, particularly the DINO series (Caron et al., 2021; Oquab et al., 2024; Siméoni et al., 2025), enriched semantic capture, and masked image modeling (He et al., 2022; Bao et al., 2022; Assran et al., 2023) showed that predicting masked tokens yields highly transferable embeddings. Vision-language models such as CLIP (Radford et al., 2021) and SigLIP (Zhai et al., 2023b) extend this by grounding visual features in text. These pre-trained representations form well-structured manifolds that benefit anomaly detection, where distinguishing normal from abnormal patterns hinges on the underlying feature space (Heckler et al., 2023). Inspired particularly by I-JEPA (Assran et al., 2023), which predicts masked tokens directly in latent space, we build on strong embedding structures and introduce a projector to map mixed anomalous and normal feature tokens back to the normal manifold.

## 5 CONCLUSION

In this work, we presented a novel few-shot anomaly detection approach, FOUNDAD, leveraging the pure visual embeddings from foundation encoders *without text-prompt assistance*. By uncovering the direct correlation between embedding distance and anomaly amount, we designed a lightweight nonlinear feature projector that efficiently maps features onto the normal image manifold. This method effectively addresses the challenge posed by limited anomaly examples, achieving robust detection with minimal training data. Extensive evaluations across multiple foundational visual encoders demonstrated that our method surpasses current few-shot anomaly detection methods. Our findings emphasize the potential of foundation visual encoders for anomaly detection, advocating broader exploration and application in industrial inspection scenarios. In future work, we plan to comprehensively investigate how diverse anomaly synthesis strategies applied before visual feature extraction affect FOUNDAD's overall learning performance.

**Reproducibility Statement**   We have made extensive efforts to ensure the reproducibility of our work. The architecture details of our framework, including the encoders and the non-linear projector, are described in section 2. Training settings, hyperparameters, and evaluation protocols are provided in section 3 and Supplementary Material A. All datasets used are publicly available. We expose some typical failure cases in Supplementary Material B to provide research insights. Additional qualitative and quantitative results, as well as studies about the Top-K selection on each dataset, are included in Supplementary Material C. Finally, we provide our code and instructions at https://github.com/ymxlzgy/FoundAD.

**LLM Usage**   We used ChatGPT solely as a writing assistant for grammar checking and language polishing of the manuscript. All research ideas, experiments, analyses, and conclusions were conceived and conducted entirely by the authors. ChatGPT and other LLMs were not used for obtaining ideas, for technical content generation, or for experimental design.

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

**Foundation Visual Encoders Are Secretly Few-Shot Anomaly Detectors**

Supplementary Material

In this material, we provide more details about the ablation of foundation encoders, illustrations of some typical failure cases, and additional quantitative and qualitative results.

## A  EXPERIMENTAL DETAIL FOR ENCODER ABLATION

We performed experiments with various encoders under a 1-shot setting on MVTec-AD. To maintain consistency with the default encoder DINOv3 of FOUNDAD, use the same sampled training instances across different encoders. The projector was uniformly maintained at a depth of 6, and the learning rate was consistently set to 0.001 for all cases.

To ensure a fair comparison, we use ViT-Base for DINOv3, DINOv2, DINO, SigLIP, and CLIP, all in the third-to-last layer. We conducted the experiment for DINOSigLIP using ViT-L due to availability. However, since different encoders require varying default image input sizes and patch sizes, the input patch numbers for the projector was adjusted accordingly: (1) For DINOv2 and DINO ViT-B, the input images were resized to 518, with a patch size of 14. (2) For DINOv3 and SigLIP ViT-B, the input images were resized to 512, with a patch size of 16. (3) For CLIP ViT-B, the default image size was 224 with a patch size of 16. (4) For DINOSigLIP, we employed the implementation from Prismatic VLMs (Karamcheti et al., 2024) [1], which combines DINOv2 ViT-L and SigLIP ViT-SO embeddings with a linear projection at an input resolution of 384. (5) For WideResNet, we adopted the methodology identical to PatchCore (Roth et al., 2022).

## B  FAILURE CASES

While FOUNDAD achieves generally strong performance in both image classification and anomaly localization, it can still be challenged under certain conditions. Typical failure cases are shown in Figure 7. In Figure 7 (a), the test screw appears in a different orientation from the training samples. Without a spatial transformation mechanism to align features as adopted in RegAD (Huang et al., 2022), FOUNDAD fails to recognize the anomaly. Similarly, a transistor with missing pins can be localized, but precise segmentation is unsuccessful. We attribute this to two factors: (i) the projector is trained solely on normal data and therefore lacks priors about the morphology of potential anomalies; and (ii) the projector, inspired by I-JEPA (Assran et al., 2023), is non-generative, limiting its ability to complete anomaly masks. In Figure 7 (b), the candle exhibits severe appearance variations caused by exposure. The anomaly becomes nearly indistinguishable from the background, hindering representation learning and degrading detection quality. Moreover, FOUNDAD occasionally misclassifies background artifacts as anomalies. For example, previously unseen stains on the platform of Pcb2 are incorrectly highlighted during inference.

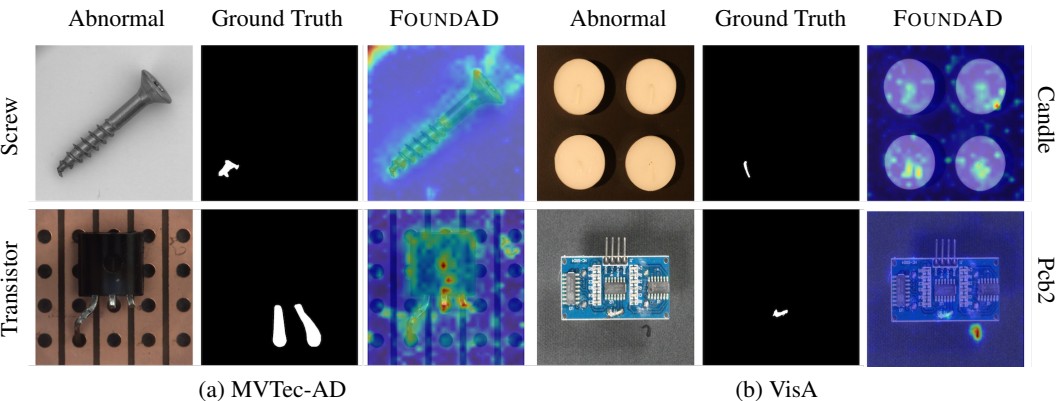

Figure 7: Typical failure cases of FOUNDAD.

---

[1] https://github.com/tri-ml/prismatic-vlms

## C  ADDITIONAL RESULTS

Table 7: Results on MVTec-AD and VisA under multiple seeds.

| Setting | Seed | MVTec-AD | | | | VisA | | | |
|---|---|---|---|---|---|---|---|---|---|
| | | I-AUROC | AUPR | P-AUROC | PRO | I-AUROC | AUPR | P-AUROC | PRO |
| 1-shot | 1 | 96.0 | 97.8 | 96.9 | 93.1 | 92.5 | 91.5 | 99.7 | 97.8 |
| | 2 | 96.0 | 97.6 | 96.9 | 92.8 | 91.9 | 91.9 | 99.7 | 98.2 |
| | 3 | 96.4 | 98.2 | 96.5 | 92.6 | 93.4 | 92.7 | 99.7 | 98.1 |
| | Avg | 96.1 | 97.9 | 96.8 | 92.8 | 92.6 | 92.0 | 99.7 | 98.0 |
| 2-shot | 1 | 96.8 | 98.3 | 97.0 | 93.3 | 93.9 | 93.2 | 99.7 | 98.1 |
| | 2 | 96.9 | 98.2 | 97.2 | 93.7 | 93.5 | 93.3 | 99.7 | 98.3 |
| | 3 | 96.6 | 98.3 | 96.9 | 92.9 | 94.0 | 93.3 | 99.7 | 98.2 |
| | Avg | 96.8 | 98.3 | 97.0 | 93.3 | 93.8 | 93.3 | 99.7 | 98.2 |
| 4-shot | 1 | 97.2 | 98.6 | 97.2 | 93.5 | 94.2 | 93.4 | 99.7 | 98.2 |
| | 2 | 97.2 | 98.6 | 97.3 | 93.7 | 94.5 | 94.6 | 99.7 | 98.4 |
| | 3 | 97.0 | 98.5 | 97.1 | 93.4 | 94.6 | 93.9 | 99.7 | 98.6 |
| | Avg | 97.1 | 98.6 | 97.2 | 93.5 | 94.4 | 94.0 | 99.7 | 98.4 |

### C.1  MULTIPLE RUN RESULTS

To demonstrate the robustness and reliability of FOUNDAD, we repeated the experiments in three random seeds on the MVTec-AD dataset. The detailed experimental results are summarized in Table 7. These results indicate that FOUNDAD consistently maintains strong generalization capability and stable anomaly detection performance even under limited data conditions.

All four metrics consistently improve as the number of shots increases, particularly on MVTec-AD, indicating that our method can effectively leverage additional supervision to enhance anomaly detection performance. As shown in Figure 8, the standard deviations are generally small across runs, demonstrating strong stability and robustness. On VisA, interestingly, the P-AUROC remains nearly saturated at 99.7 across all settings, showing that the method performs highly reliably at the pixel level. I-AUROC and AUPR exhibit steeper gains on MVTec-AD, likely due to its more diverse visual patterns, where additional examples provide more substantial benefit, whereas VisA benefits from more homogeneous object categories. Although the PRO metric shows slight fluctuations, it consistently stays high, underscoring the method's effectiveness in localizing anomalies with pixel-level precision. Overall, the results demonstrate our method's robustness, consistency, and scalability in few-shot settings, making it well-suited for real-world industrial scenarios with limited annotations.

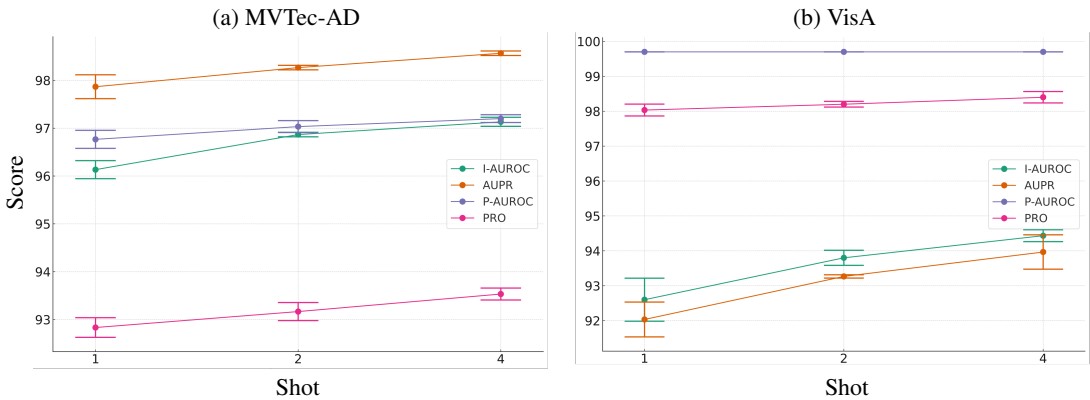

Figure 8: Mean and standard deviation of four evaluation metrics across 1-, 2-, and 4-shot settings on MVTec-AD and VisA.

### C.2  ADDITIONAL QUALITATIVE COMPARISONS

We provide several visual comparisons against LogSAD (Zhang et al., 2025b) in Figure 9. It clearly shows that FOUNDAD localizes anomaly precisely, while producing significantly less noise.

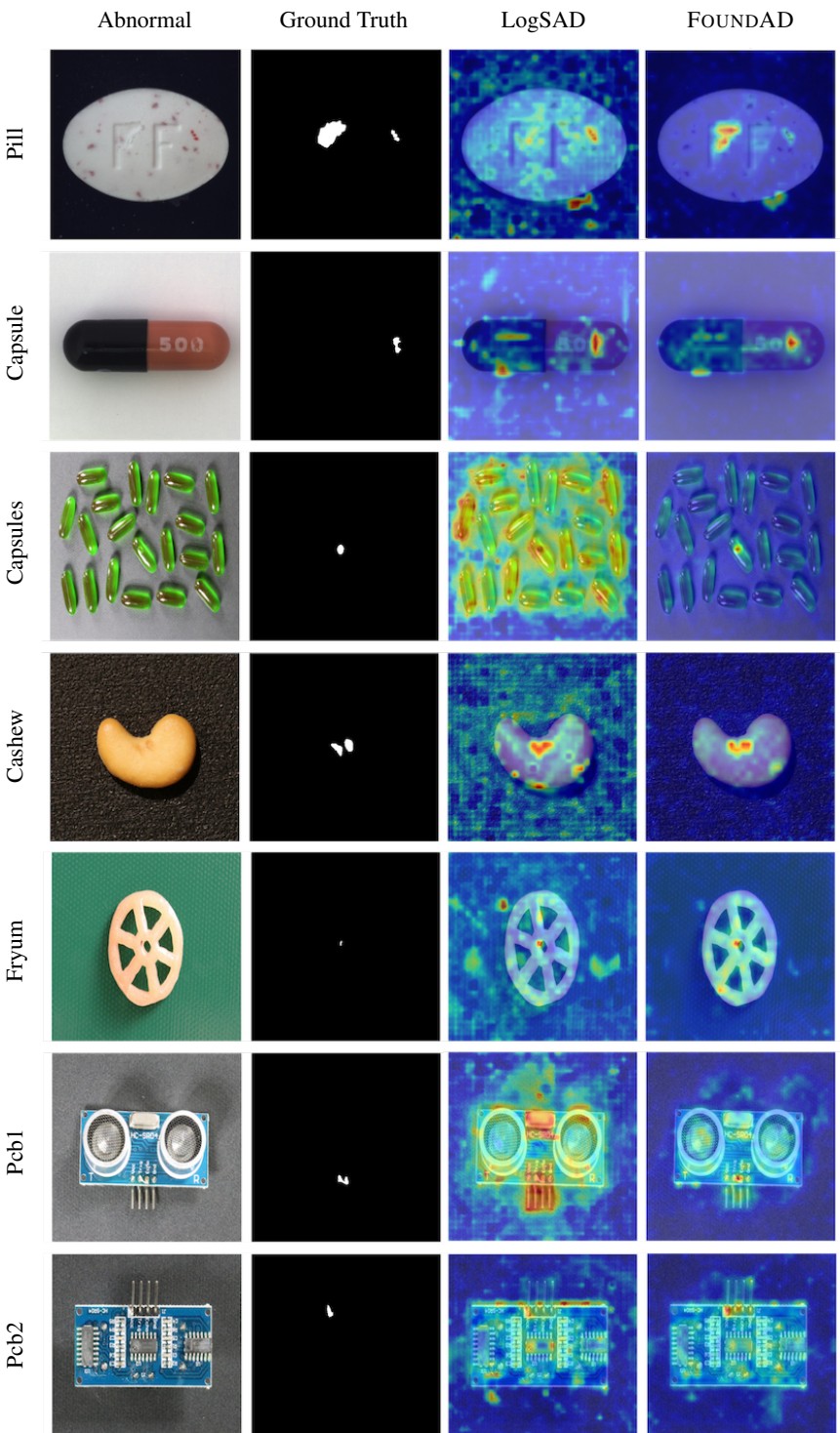

Figure 9: Visual comparisons with LogSAD (Zhang et al., 2025b).

## C.3 CLASS-WISE QUANTITATIVE RESULTS

We report the class-wise quantitative results of FOUNDAD with DINOv3 under 1, 2, and 4 shot settings on MVTec-AD and VisA in detail in Table 8 and Table 9.

Table 8: Results on MVTec-AD for 1-shot, 2-shot, and 4-shot. Metrics include I-AUROC (%), AUPR (%), and pixel-level P-AUROC (%), PRO (%).

| Class | 1-shot | | | | 2-shot | | | | 4-shot | | | |
|---|---|---|---|---|---|---|---|---|---|---|---|---|
| | I-AUROC | AUPR | P-AUROC | PRO | I-AUROC | AUPR | P-AUROC | PRO | I-AUROC | AUPR | P-AUROC | PRO |
| Bottle | 100.0 | 100.0 | 98.6 | 96.2 | 100.0 | 100.0 | 98.6 | 95.8 | 100.0 | 100.0 | 98.6 | 95.9 |
| Cable | 89.6 | 94.1 | 94.7 | 89.2 | 92.3 | 96.3 | 95.0 | 89.9 | 92.4 | 96.4 | 94.7 | 89.6 |
| Capsule | 89.5 | 97.5 | 98.6 | 94.3 | 89.6 | 97.6 | 98.8 | 94.9 | 90.1 | 97.7 | 98.8 | 94.8 |
| Carpet | 100.0 | 100.0 | 99.5 | 98.3 | 100.0 | 100.0 | 99.4 | 98.3 | 100.0 | 100.0 | 99.5 | 98.2 |
| Grid | 99.8 | 99.9 | 99.2 | 96.1 | 99.9 | 99.9 | 99.3 | 96.6 | 100.0 | 100.0 | 99.3 | 96.8 |
| Hazelnut | 99.3 | 99.7 | 99.5 | 93.0 | 99.9 | 99.9 | 99.5 | 93.2 | 99.8 | 99.8 | 99.5 | 93.4 |
| Leather | 100.0 | 100.0 | 99.4 | 98.7 | 100.0 | 100.0 | 99.4 | 98.7 | 100.0 | 100.0 | 99.4 | 98.4 |
| Metal_nut | 99.8 | 100.0 | 92.5 | 91.8 | 99.9 | 100.0 | 94.3 | 93.3 | 100.0 | 100.0 | 96.3 | 95.3 |
| Pill | 97.7 | 99.6 | 94.5 | 97.0 | 97.8 | 99.6 | 94.6 | 97.1 | 98.1 | 99.7 | 95.0 | 96.7 |
| Screw | 84.8 | 94.3 | 98.4 | 92.6 | 87.6 | 95.2 | 98.7 | 92.8 | 89.0 | 95.7 | 98.7 | 93.3 |
| Tile | 100.0 | 100.0 | 97.2 | 95.4 | 100.0 | 100.0 | 97.3 | 95.1 | 100.0 | 100.0 | 97.4 | 94.9 |
| Toothbrush | 99.8 | 99.9 | 99.2 | 94.1 | 99.2 | 99.7 | 99.2 | 95.2 | 100.0 | 100.0 | 99.4 | 95.6 |
| Transistor | 86.3 | 85.0 | 85.0 | 63.3 | 89.2 | 86.6 | 86.6 | 63.8 | 92.5 | 90.0 | 87.4 | 65.7 |
| Wood | 95.4 | 97.9 | 96.2 | 96.0 | 96.9 | 98.7 | 96.2 | 95.9 | 95.2 | 98.6 | 96.2 | 96.1 |
| Zipper | 99.8 | 99.9 | 99.0 | 96.8 | 99.8 | 99.9 | 99.0 | 97.1 | 99.9 | 100.0 | 99.1 | 97.0 |
| **Average** | 96.1 | 97.9 | 96.8 | 92.8 | 96.8 | 98.3 | 97.0 | 93.3 | 97.1 | 98.6 | 97.2 | 93.5 |

Table 9: Results of FOUNDAD on VisA for 1-shot, 2-shot, and 4-shot. Metrics include I-AUROC (%), AUPR (%) and pixel-level P-AUROC (%), PRO (%).

| Class | 1-shot | | | | 2-shot | | | | 4-shot | | | |
|---|---|---|---|---|---|---|---|---|---|---|---|---|
| | I-AUROC | AUPR | P-AUROC | PRO | I-AUROC | AUPR | P-AUROC | PRO | I-AUROC | AUPR | P-AUROC | PRO |
| Candle | 94.4 | 93.2 | 99.5 | 98.4 | 94.7 | 93.3 | 99.6 | 98.7 | 94.3 | 94.1 | 99.5 | 98.4 |
| Capsules | 98.9 | 99.3 | 99.8 | 99.0 | 99.0 | 99.4 | 99.8 | 99.0 | 99.1 | 99.4 | 99.8 | 99.0 |
| Cashew | 97.4 | 98.6 | 99.6 | 98.8 | 97.0 | 98.5 | 99.6 | 98.5 | 97.6 | 98.7 | 99.6 | 98.5 |
| Chewinggum | 98.0 | 99.1 | 99.9 | 99.0 | 98.5 | 99.3 | 99.9 | 99.1 | 98.5 | 99.3 | 99.8 | 98.9 |
| Fryum | 96.8 | 98.5 | 99.8 | 98.0 | 96.8 | 98.5 | 99.8 | 98.2 | 97.2 | 98.6 | 99.7 | 98.1 |
| Macaroni1 | 93.1 | 92.2 | 99.5 | 98.0 | 93.8 | 92.9 | 99.6 | 98.3 | 94.0 | 93.0 | 99.6 | 98.3 |
| Macaroni2 | 72.3 | 64.9 | 99.9 | 99.8 | 75.2 | 68.0 | 99.9 | 99.8 | 77.2 | 71.1 | 99.9 | 99.9 |
| Pcb1 | 94.1 | 90.8 | 99.5 | 98.1 | 95.3 | 92.9 | 99.5 | 98.2 | 96.9 | 94.6 | 99.6 | 98.5 |
| Pcb2 | 92.6 | 90.5 | 99.6 | 97.6 | 93.9 | 92.1 | 99.6 | 97.6 | 93.9 | 91.8 | 99.7 | 98.0 |
| Pcb3 | 82.0 | 84.7 | 99.8 | 99.1 | 87.8 | 90.0 | 99.9 | 99.4 | 90.5 | 91.6 | 99.9 | 99.5 |
| Pcb4 | 96.6 | 94.8 | 99.7 | 93.2 | 97.2 | 96.6 | 99.7 | 94.7 | 97.5 | 96.5 | 99.7 | 95.2 |
| Pipe_fryum | 95.0 | 97.5 | 99.7 | 96.9 | 96.1 | 98.0 | 99.7 | 97.0 | 96.9 | 98.5 | 99.7 | 96.8 |
| **Average** | 92.6 | 92.0 | 99.7 | 98.0 | 93.8 | 93.3 | 99.7 | 98.2 | 94.4 | 94.0 | 99.7 | 98.4 |

## C.4 CLASS-WISE COMPARISON TO OTHER METHODS

The class-wise comparisons are presented with I-AUROC in Table 10-Table 11, AUPR in Table 12-Table 13, P-AUROC in Table 14-Table 15, and PRO in Table 16-Table 17.

Table 10: I-AUROC (%) results of MVTec-AD for 1-shot, 2-shot, and 4-shot.

| Class | 1-shot | | | | 2-shot | | | | 4-shot | | | |
|---|---|---|---|---|---|---|---|---|---|---|---|---|
| | WinCLIP | PromptAD | IIPAD | Ours | WinCLIP | PromptAD | IIPAD | Ours | WinCLIP | PromptAD | IIPAD | Ours |
| Bottle | 98.9 | 98.6 | 99.7 | 100.0 | 99.2 | 100.0 | 99.8 | 100.0 | 99.2 | 99.0 | 99.1 | 99.1 |
| Cable | 78.0 | 83.6 | 92.8 | 89.6 | 83.9 | 87.2 | 92.1 | 92.3 | 82.3 | 88.7 | 95.4 | 92.4 |
| Capsule | 75.5 | 64.2 | 80.5 | 89.5 | 65.5 | 65.3 | 91.8 | 89.6 | 80.1 | 93.4 | 94.5 | 90.1 |
| Carpet | 99.9 | 100.0 | 100.0 | 100.0 | 99.9 | 100.0 | 100.0 | 100.0 | 99.9 | 100.0 | 100.0 | 100.0 |
| Grid | 99.6 | 98.8 | 97.0 | 99.8 | 99.2 | 97.4 | 97.0 | 99.9 | 99.5 | 100.0 | 96.0 | 100.0 |
| Hazelnut | 94.9 | 98.4 | 98.0 | 99.3 | 95.2 | 99.8 | 98.5 | 99.9 | 94.7 | 99.0 | 98.5 | 99.8 |
| Leather | 100.0 | 100.0 | 100.0 | 100.0 | 100.0 | 100.0 | 100.0 | 100.0 | 100.0 | 100.0 | 100.0 | 100.0 |
| Metal_nut | 98.0 | 97.6 | 99.4 | 99.8 | 97.8 | 96.2 | 99.7 | 99.9 | 98.9 | 100.0 | 99.9 | 100.0 |
| Pill | 88.9 | 87.9 | 96.6 | 97.7 | 91.8 | 89.1 | 96.0 | 97.8 | 91.1 | 90.4 | 96.6 | 98.1 |
| Screw | 85.1 | 74.0 | 76.8 | 84.8 | 82.7 | 81.2 | 81.5 | 87.6 | 84.4 | 84.2 | 82.1 | 89.0 |
| Tile | 100.0 | 99.8 | 99.7 | 100.0 | 100.0 | 99.3 | 99.5 | 100.0 | 100.0 | 99.2 | 99.9 | 100.0 |
| Toothbrush | 94.2 | 94.4 | 91.9 | 99.8 | 93.9 | 100.0 | 92.5 | 99.2 | 98.1 | 98.8 | 92.5 | 100.0 |
| Transistor | 85.5 | 73.7 | 91.4 | 86.3 | 85.4 | 87.2 | 90.4 | 89.2 | 85.6 | 94.4 | 91.2 | 92.5 |
| Wood | 98.7 | 98.6 | 99.4 | 95.4 | 98.9 | 98.9 | 99.2 | 96.9 | 98.9 | 99.2 | 99.6 | 95.2 |
| Zipper | 94.9 | 95.3 | 89.4 | 99.8 | 97.2 | 93.5 | 95.5 | 99.8 | 97.0 | 95.8 | 96.0 | 99.9 |
| **Mean** | 92.8 | 86.3 | 94.2 | 96.1 | 92.7 | 89.2 | 95.6 | 96.8 | 94.0 | 90.6 | 96.1 | 97.1 |

Table 11: I-AUROC (%) results of VisA for 1-shot, 2-shot, and 4-shot.

| Class | 1-shot | | | | 2-shot | | | | 4-shot | | | |
|---|---|---|---|---|---|---|---|---|---|---|---|---|
| | WinCLIP | PromptAD | IIPAD | Ours | WinCLIP | PromptAD | IIPAD | Ours | WinCLIP | PromptAD | IIPAD | Ours |
| Candle | 96.3 | 91.8 | 91.9 | 94.4 | 96.4 | 92.0 | 95.5 | 94.7 | 96.9 | 92.9 | 95.9 | 94.3 |
| Capsules | 79.3 | 83.2 | 88.9 | 98.9 | 81.6 | 78.7 | 90.3 | 99.0 | 83.0 | 81.7 | 90.5 | 99.1 |
| Cashew | 93.9 | 88.9 | 85.6 | 97.4 | 92.6 | 89.6 | 86.7 | 97.0 | 92.6 | 88.0 | 91.2 | 97.6 |
| Chewinggum | 97.9 | 97.3 | 97.7 | 98.0 | 98.1 | 97.1 | 97.8 | 98.5 | 98.4 | 98.1 | 98.0 | 98.5 |
| Fryum | 92.8 | 88.0 | 89.9 | 96.8 | 90.1 | 85.7 | 92.7 | 96.8 | 91.6 | 90.6 | 93.3 | 97.2 |
| Macaroni1 | 81.9 | 87.3 | 85.1 | 93.1 | 86.4 | 87.4 | 84.7 | 93.8 | 86.9 | 89.1 | 88.4 | 94.0 |
| Macaroni2 | 78.1 | 60.8 | 75.5 | 72.3 | 76.8 | 74.9 | 76.1 | 75.2 | 79.0 | 80.5 | 78.1 | 77.2 |
| Pcb1 | 83.8 | 83.0 | 83.5 | 94.1 | 85.5 | 82.9 | 86.5 | 95.3 | 86.0 | 86.1 | 85.2 | 96.9 |
| Pcb2 | 58.4 | 77.9 | 72.6 | 92.6 | 56.8 | 84.4 | 75.2 | 93.9 | 59.4 | 81.1 | 75.5 | 93.9 |
| Pcb3 | 64.9 | 79.9 | 71.8 | 82.0 | 67.7 | 71.7 | 70.7 | 87.8 | 65.6 | 87.1 | 74.7 | 90.5 |
| Pcb4 | 72.1 | 96.5 | 82.9 | 96.6 | 73.6 | 96.0 | 84.4 | 97.2 | 70.7 | 85.3 | 88.7 | 97.5 |
| Pipefryum | 98.2 | 98.9 | 99.8 | 95.0 | 98.5 | 99.6 | 99.9 | 96.1 | 98.4 | 99.3 | 99.8 | 96.9 |
| **Mean** | 83.1 | 80.8 | 85.4 | 92.6 | 83.7 | 84.3 | 86.7 | 93.8 | 84.1 | 85.7 | 88.3 | 94.4 |

Table 12: AUPR (%) results of MVTec-AD for 1-shot, 2-shot, and 4-shot.

| Class | 1-shot | | | | 2-shot | | | | 4-shot | | | |
|---|---|---|---|---|---|---|---|---|---|---|---|---|
| | WinCLIP | PromptAD | IIPAD | Ours | WinCLIP | PromptAD | IIPAD | Ours | WinCLIP | PromptAD | IIPAD | Ours |
| Bottle | 99.7 | 99.6 | 99.9 | 100.0 | 99.8 | 100.0 | 100.0 | 100.0 | 99.8 | 99.7 | 99.7 | 100.0 |
| Cable | 87.1 | 91.2 | 96.1 | 94.1 | 90.6 | 92.8 | 95.8 | 96.3 | 90.0 | 93.6 | 97.5 | 96.4 |
| Capsule | 93.9 | 85.7 | 95.4 | 97.5 | 88.3 | 85.5 | 98.3 | 97.6 | 94.8 | 98.5 | 98.9 | 97.7 |
| Carpet | 100.0 | 100.0 | 100.0 | 100.0 | 100.0 | 100.0 | 100.0 | 100.0 | 100.0 | 100.0 | 100.0 | 100.0 |
| Grid | 99.9 | 99.5 | 99.0 | 99.9 | 99.7 | 99.1 | 99.0 | 99.9 | 99.8 | 100.0 | 98.6 | 100.0 |
| Hazelnut | 97.4 | 99.1 | 99.2 | 99.7 | 97.5 | 99.9 | 99.4 | 99.9 | 97.2 | 99.4 | 99.4 | 99.8 |
| Leather | 100.0 | 100.0 | 100.0 | 100.0 | 100.0 | 100.0 | 100.0 | 100.0 | 100.0 | 100.0 | 100.0 | 100.0 |
| Metal_nut | 99.6 | 99.3 | 99.9 | 100.0 | 99.5 | 98.3 | 99.9 | 100.0 | 99.8 | 100.0 | 100.0 | 100.0 |
| Pill | 97.6 | 96.8 | 99.4 | 99.6 | 98.3 | 97.1 | 99.3 | 99.6 | 98.2 | 97.7 | 99.4 | 99.7 |
| Screw | 95.1 | 91.19 | 87.0 | 94.3 | 93.3 | 93.5 | 92.6 | 95.2 | 94.0 | 93.7 | 92.0 | 95.7 |
| Tile | 100.0 | 99.9 | 99.9 | 100.0 | 100.0 | 99.7 | 99.8 | 100.0 | 100.0 | 99.6 | 99.9 | 100.0 |
| Toothbrush | 97.7 | 97.7 | 97.3 | 99.9 | 97.6 | 100.0 | 97.5 | 99.7 | 99.3 | 99.5 | 97.5 | 100.0 |
| Transistor | 80.8 | 62.2 | 89.8 | 85.0 | 81.0 | 77.2 | 88.3 | 86.6 | 82.6 | 92.2 | 89.2 | 90.0 |
| Wood | 99.6 | 99.5 | 99.8 | 97.9 | 99.7 | 99.6 | 99.8 | 98.7 | 99.7 | 99.7 | 99.9 | 98.6 |
| Zipper | 98.6 | 98.8 | 95.6 | 99.9 | 99.2 | 98.2 | 98.7 | 99.9 | 99.2 | 98.9 | 98.8 | 100.0 |
| **Mean** | 96.5 | 93.4 | 97.2 | 97.9 | 96.3 | 94.8 | 97.9 | 98.3 | 97.0 | 96.5 | 98.1 | 98.6 |

Table 13: AUPR (%) results of VisA for 1-shot, 2-shot, and 4-shot.

| Class | 1-shot | | | | 2-shot | | | | 4-shot | | | |
|---|---|---|---|---|---|---|---|---|---|---|---|---|
| | WinCLIP | PromptAD | IIPAD | Ours | WinCLIP | PromptAD | IIPAD | Ours | WinCLIP | PromptAD | IIPAD | Ours |
| Candle | 96.7 | 90.7 | 94.1 | 93.2 | 96.9 | 91.7 | 95.3 | 93.3 | 97.3 | 92.8 | 95.5 | 94.1 |
| Capsules | 87.0 | 90.0 | 94.7 | 99.3 | 89.1 | 86.7 | 94.9 | 99.4 | 90.0 | 89.0 | 96.0 | 99.4 |
| Cashew | 97.4 | 95.0 | 95.7 | 98.6 | 96.7 | 95.1 | 95.2 | 98.5 | 96.8 | 94.7 | 96.0 | 98.7 |
| Chewinggum | 99.1 | 98.9 | 99.1 | 99.1 | 99.2 | 98.8 | 99.2 | 99.3 | 99.3 | 99.2 | 99.3 | 99.3 |
| Fryum | 96.9 | 94.6 | 96.3 | 98.5 | 95.4 | 93.9 | 96.8 | 98.5 | 96.1 | 95.9 | 97.3 | 98.6 |
| Macaroni1 | 82.8 | 89.7 | 89.3 | 92.2 | 87.2 | 89.0 | 90.7 | 92.9 | 87.4 | 91.1 | 91.2 | 93.0 |
| Macaroni2 | 80.1 | 61.5 | 79.3 | 64.9 | 79.0 | 78.2 | 80.2 | 68.0 | 81.9 | 81.2 | 80.6 | 71.1 |
| Pcb1 | 83.7 | 77.3 | 77.4 | 90.8 | 84.1 | 79.0 | 78.5 | 92.9 | 85.6 | 81.2 | 78.6 | 94.6 |
| Pcb2 | 58.6 | 79.3 | 70.2 | 90.5 | 54.6 | 85.5 | 73.6 | 92.1 | 61.3 | 80.8 | 73.9 | 91.8 |
| Pcb3 | 66.2 | 81.6 | 73.2 | 84.7 | 67.3 | 73.5 | 74.3 | 90.0 | 64.6 | 88.0 | 74.6 | 91.6 |
| Pcb4 | 73.8 | 96.1 | 80.9 | 94.8 | 70.2 | 94.8 | 81.7 | 96.6 | 73.5 | 83.6 | 84.5 | 96.5 |
| Pipefryum | 99.2 | 99.6 | 99.9 | 97.5 | 99.3 | 99.7 | 100 | 98.0 | 99.3 | 99.7 | 99.9 | 98.5 |
| **Mean** | 85.1 | 83.2 | 87.5 | 92.0 | 84.9 | 87.8 | 88.6 | 93.3 | 86.1 | 88.8 | 89.6 | 94.0 |

Table 14: P-AUROC (%) results of MVTec-AD for 1-shot, 2-shot, and 4-shot.

| Class | 1-shot | | | | 2-shot | | | | 4-shot | | | |
|---|---|---|---|---|---|---|---|---|---|---|---|---|
| | WinCLIP | PromptAD | IIPAD | Ours | WinCLIP | PromptAD | IIPAD | Ours | WinCLIP | PromptAD | IIPAD | Ours |
| Bottle | 95.1 | 94.5 | 98.4 | 98.6 | 95.5 | 95.9 | 98.6 | 98.6 | 95.2 | 96.9 | 98.6 | 98.6 |
| Cable | 74.6 | 76.8 | 94.9 | 94.7 | 76.4 | 80.7 | 95.3 | 95.0 | 77.1 | 83.7 | 96.4 | 94.7 |
| Capsule | 95.6 | 94.6 | 97.2 | 98.6 | 94.7 | 94.8 | 97.5 | 98.8 | 96.3 | 97.7 | 97.2 | 98.8 |
| Carpet | 99.1 | 99.1 | 99.5 | 99.5 | 99.1 | 99.2 | 99.5 | 99.4 | 99.1 | 99.2 | 99.5 | 99.5 |
| Grid | 95.1 | 96.8 | 96.6 | 99.2 | 96.3 | 96.0 | 96.3 | 99.3 | 96.0 | 96.8 | 98.1 | 99.3 |
| Hazelnut | 98.6 | 96.9 | 98.4 | 99.5 | 98.7 | 98.0 | 98.7 | 99.5 | 98.7 | 98.1 | 98.8 | 99.5 |
| Leather | 99.3 | 99.3 | 99.4 | 99.4 | 99.3 | 99.4 | 99.3 | 99.4 | 99.4 | 99.4 | 99.5 | 99.4 |
| Metal_nut | 77.9 | 94.2 | 94.4 | 92.5 | 76.1 | 94.8 | 94.3 | 94.3 | 79.5 | 93.2 | 94.8 | 96.3 |
| Pill | 93.9 | 92.3 | 96.6 | 94.5 | 94.1 | 94.3 | 97.0 | 94.6 | 94.4 | 95.3 | 96.9 | 95.0 |
| Screw | 96.7 | 95.7 | 96.0 | 98.4 | 97.0 | 96.2 | 96.5 | 98.7 | 96.5 | 97.2 | 96.9 | 98.7 |
| Tile | 92.2 | 94.3 | 96.9 | 97.2 | 92.6 | 95.5 | 97.1 | 97.3 | 92.2 | 95.9 | 97.3 | 97.4 |
| Toothbrush | 95.1 | 99.0 | 97.4 | 99.2 | 84.5 | 99.2 | 97.4 | 99.2 | 96.8 | 99.2 | 97.9 | 99.4 |
| Transistor | 85.1 | 75.5 | 86.6 | 85.0 | 95.3 | 84.4 | 88.4 | 86.6 | 84.4 | 87.5 | 89.6 | 87.4 |
| Wood | 94.7 | 95.8 | 97.0 | 96.2 | 94.7 | 95.3 | 97.3 | 96.2 | 94.6 | 95.6 | 97.4 | 96.2 |
| Zipper | 92.9 | 97.6 | 96.0 | 99.0 | 92.4 | 97.3 | 96.7 | 99.0 | 92.9 | 96.9 | 96.7 | 99.1 |
| **Mean** | 92.4 | 91.8 | 96.4 | 96.8 | 92.4 | 92.2 | 96.7 | 97.0 | 92.9 | 92.9 | 97.0 | 97.2 |

Table 15: P-AUROC (%) results of VisA for 1-shot, 2-shot, and 4-shot.

| Class | 1-shot | | | | 2-shot | | | | 4-shot | | | |
|---|---|---|---|---|---|---|---|---|---|---|---|---|
| | WinCLIP | PromptAD | IIPAD | Ours | WinCLIP | PromptAD | IIPAD | Ours | WinCLIP | PromptAD | IIPAD | Ours |
| Candle | 93.8 | 97.1 | 98.5 | 99.5 | 94.7 | 97.7 | 98.5 | 99.6 | 95.0 | 97.7 | 98.6 | 99.5 |
| Capsules | 93.2 | 96.7 | 97.1 | 99.8 | 93.0 | 97.2 | 97.3 | 99.8 | 93.2 | 97.4 | 97.9 | 99.8 |
| Cashew | 94.6 | 97.9 | 97.6 | 99.6 | 95.3 | 97.8 | 97.1 | 99.6 | 94.7 | 97.9 | 97.5 | 99.6 |
| Chewinggum | 98.9 | 99.2 | 99.6 | 99.9 | 98.9 | 99.0 | 99.5 | 99.8 | 98.9 | 99.1 | 99.5 | 99.8 |
| Fryum | 95.1 | 95.6 | 96.2 | 99.8 | 95.6 | 95.8 | 96.1 | 99.8 | 95.4 | 96.0 | 95.8 | 99.7 |
| Macaroni1 | 95.6 | 97.6 | 97.5 | 99.5 | 96.7 | 98.8 | 97.6 | 99.6 | 97.0 | 98.6 | 97.8 | 99.6 |
| Macaroni2 | 94.0 | 95.6 | 96.6 | 99.9 | 94.4 | 96.1 | 96.9 | 99.9 | 93.8 | 98.1 | 96.9 | 99.9 |
| Pcb1 | 94.1 | 96.9 | 98.5 | 99.5 | 94.6 | 98.1 | 98.5 | 99.5 | 94.7 | 98.8 | 98.7 | 99.6 |
| Pcb2 | 92.4 | 94.0 | 94.6 | 99.6 | 93.1 | 95.1 | 96.8 | 99.6 | 93.3 | 95.6 | 96.6 | 99.7 |
| Pcb3 | 91.6 | 95.3 | 93.0 | 99.8 | 92.4 | 95.5 | 93.7 | 99.9 | 93.2 | 96.4 | 94.0 | 99.9 |
| Pcb4 | 94.2 | 95.7 | 95.2 | 99.7 | 94.9 | 96.8 | 96.6 | 99.7 | 95.6 | 96.9 | 97.2 | 99.7 |
| Pipefryum | 97.9 | 98.6 | 98.3 | 99.7 | 97.8 | 98.8 | 98.5 | 99.7 | 97.8 | 98.9 | 98.5 | 99.7 |
| **Mean** | 94.6 | 96.3 | 96.9 | 99.7 | 95.1 | 96.9 | 97.2 | 99.7 | 95.2 | 97.2 | 97.4 | 99.7 |

Table 16: PRO (%) results of MVTec-AD for 1-shot, 2-shot, and 4-shot.

| Class | 1-shot | | | | 2-shot | | | | 4-shot | | | |
|---|---|---|---|---|---|---|---|---|---|---|---|---|
| | WinCLIP | PromptAD | IIPAD | Ours | WinCLIP | PromptAD | IIPAD | Ours | WinCLIP | PromptAD | IIPAD | Ours |
| Bottle | 84.5 | 89.1 | 94.9 | 96.2 | 85.8 | 91.6 | 95.2 | 95.8 | 84.8 | 93.4 | 95.5 | 95.9 |
| Cable | 55.9 | 70.4 | 86.7 | 89.2 | 62.1 | 71.2 | 89.5 | 89.9 | 61.5 | 75.8 | 91.2 | 89.6 |
| Capsule | 89.0 | 81.2 | 91.3 | 94.3 | 84.2 | 82.0 | 92.6 | 94.9 | 88.9 | 90.7 | 88.9 | 94.8 |
| Carpet | 96.4 | 96.8 | 98.1 | 98.3 | 96.1 | 96.9 | 98.0 | 98.3 | 96.0 | 96.6 | 97.7 | 98.2 |
| Grid | 85.3 | 91.9 | 88.4 | 96.1 | 88.1 | 89.2 | 87.7 | 96.6 | 86.8 | 91.9 | 91.3 | 96.8 |
| Hazelnut | 92.5 | 90.9 | 93.8 | 93.0 | 93.1 | 93.6 | 94.6 | 93.2 | 92.4 | 93.3 | 95.8 | 93.4 |
| Leather | 98.2 | 98.3 | 98.7 | 98.7 | 98.3 | 98.8 | 98.5 | 98.7 | 98.2 | 97.8 | 98.6 | 98.4 |
| Metal_nut | 77.0 | 90.1 | 91.8 | 91.8 | 75.6 | 90.7 | 92.3 | 94.3 | 79.5 | 89.8 | 93.4 | 95.3 |
| Pill | 88.9 | 90.1 | 94.8 | 97.0 | 89.6 | 93.3 | 94.1 | 97.1 | 89.9 | 94.6 | 95.6 | 96.7 |
| Screw | 87.0 | 83.9 | 85.3 | 92.6 | 88.1 | 84.5 | 85.8 | 92.8 | 86.6 | 89.1 | 87.5 | 93.3 |
| Tile | 78.7 | 89.8 | 90.4 | 95.4 | 79.7 | 90.7 | 90.7 | 95.1 | 78.2 | 91.4 | 91.2 | 94.9 |
| Toothbrush | 86.5 | 93.4 | 82.4 | 94.1 | 85.6 | 93.1 | 82.9 | 95.2 | 88.5 | 92.3 | 84.4 | 95.6 |
| Transistor | 63.5 | 59.4 | 67.4 | 63.3 | 62.8 | 66.0 | 68.6 | 63.8 | 62.6 | 69.4 | 71.2 | 65.7 |
| Wood | 86.9 | 92.8 | 94.2 | 96.0 | 87.8 | 93.4 | 94.6 | 95.9 | 88.3 | 93.1 | 94.7 | 96.1 |
| Zipper | 81.9 | 92.5 | 88.3 | 96.8 | 81.8 | 91.7 | 89.6 | 97.1 | 83.0 | 91.2 | 90.8 | 97.0 |
| **Mean** | 83.5 | 83.6 | 89.8 | 92.8 | 83.9 | 84.3 | 90.3 | 93.3 | 84.4 | 84.7 | 91.2 | 93.5 |

Table 17: PRO (%) results of VisA for 1-shot, 2-shot, and 4-shot.

| Class | 1-shot | | | | 2-shot | | | | 4-shot | | | |
|---|---|---|---|---|---|---|---|---|---|---|---|---|
| | WinCLIP | PromptAD | IIPAD | Ours | WinCLIP | PromptAD | IIPAD | Ours | WinCLIP | PromptAD | IIPAD | Ours |
| Candle | 89.6 | 92.3 | 95.0 | 98.4 | 90.2 | 92.3 | 95.3 | 98.7 | 90.5 | 92.6 | 95.4 | 98.4 |
| Capsules | 62.1 | 82.7 | 83.6 | 99.0 | 61.8 | 82.1 | 83.9 | 99.0 | 61.9 | 77.0 | 85.2 | 99.0 |
| Cashew | 87.6 | 89.9 | 92.9 | 98.8 | 86.7 | 88.1 | 93.0 | 98.5 | 86.7 | 88.3 | 92.8 | 98.5 |
| Chewinggum | 82.7 | 84.9 | 92.5 | 99.0 | 83.0 | 84.1 | 93.4 | 99.1 | 82.7 | 83.2 | 93.7 | 98.9 |
| Fryum | 87.5 | 81.9 | 87.7 | 98.0 | 87.8 | 80.8 | 88.1 | 98.2 | 88.7 | 81.9 | 89.2 | 98.1 |
| Macaroni1 | 85.6 | 88.6 | 89.3 | 98.0 | 89.8 | 90.8 | 90.1 | 98.3 | 90.1 | 93.5 | 89.9 | 98.3 |
| Macaroni2 | 81.0 | 83.7 | 86.6 | 99.8 | 81.0 | 85.2 | 86.3 | 99.8 | 79.8 | 91.2 | 87.1 | 99.9 |
| Pcb1 | 68.8 | 87.9 | 84.3 | 98.1 | 70.2 | 86.5 | 85.1 | 98.2 | 70.5 | 87.1 | 85.6 | 98.5 |
| Pcb2 | 73.6 | 73.4 | 77.3 | 97.6 | 74.0 | 76.8 | 77.8 | 97.6 | 74.1 | 77.9 | 77.4 | 98.0 |
| Pcb3 | 76.7 | 79.0 | 76.8 | 99.1 | 79.4 | 79.5 | 78.7 | 99.4 | 80.3 | 83.6 | 78.1 | 99.5 |
| Pcb4 | 79.9 | 76.7 | 83.7 | 93.2 | 82.4 | 83.7 | 84.3 | 94.7 | 83.8 | 82.0 | 87.3 | 95.2 |
| Pipe_fryum | 95.7 | 96.2 | 97.2 | 96.9 | 95.7 | 96.9 | 97.1 | 97.0 | 95.8 | 96.7 | 97.1 | 96.8 |
| **Mean** | 80.9 | 82.2 | 87.3 | 98.0 | 81.8 | 85.2 | 87.9 | 98.2 | 82.1 | 84.7 | 88.3 | 98.4 |

## C.5 CLASS-WISE QUALITATIVE RESULTS

The class-wise qualitative results of our method in 1-shot setting on MVTec-AD and VisA are shown in Figure 10, Figure 11, Figure 12, and Figure 13. We show each training sample in the first column.

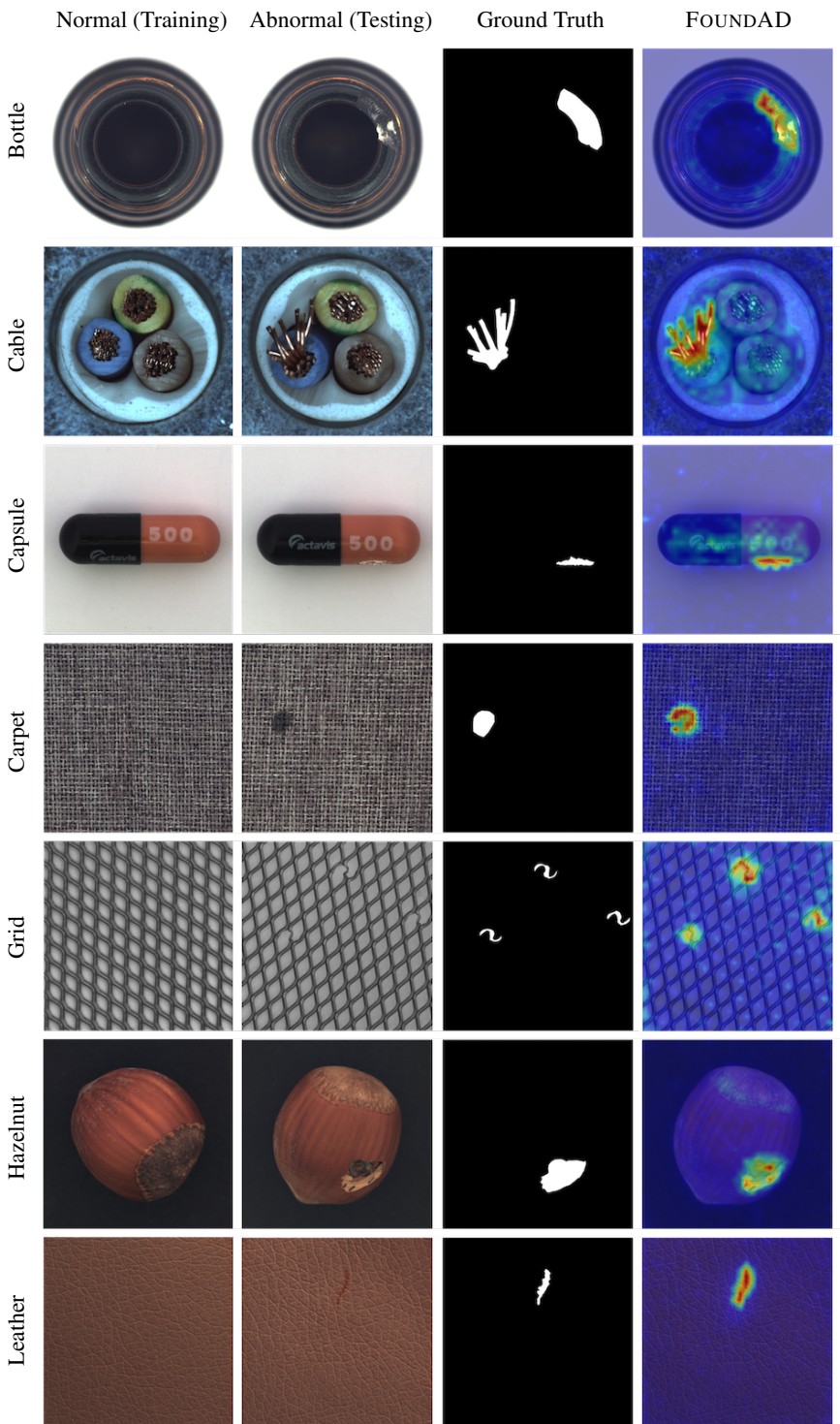

Figure 10: More Qualitative 1-shot Results on MVTec-AD (i).

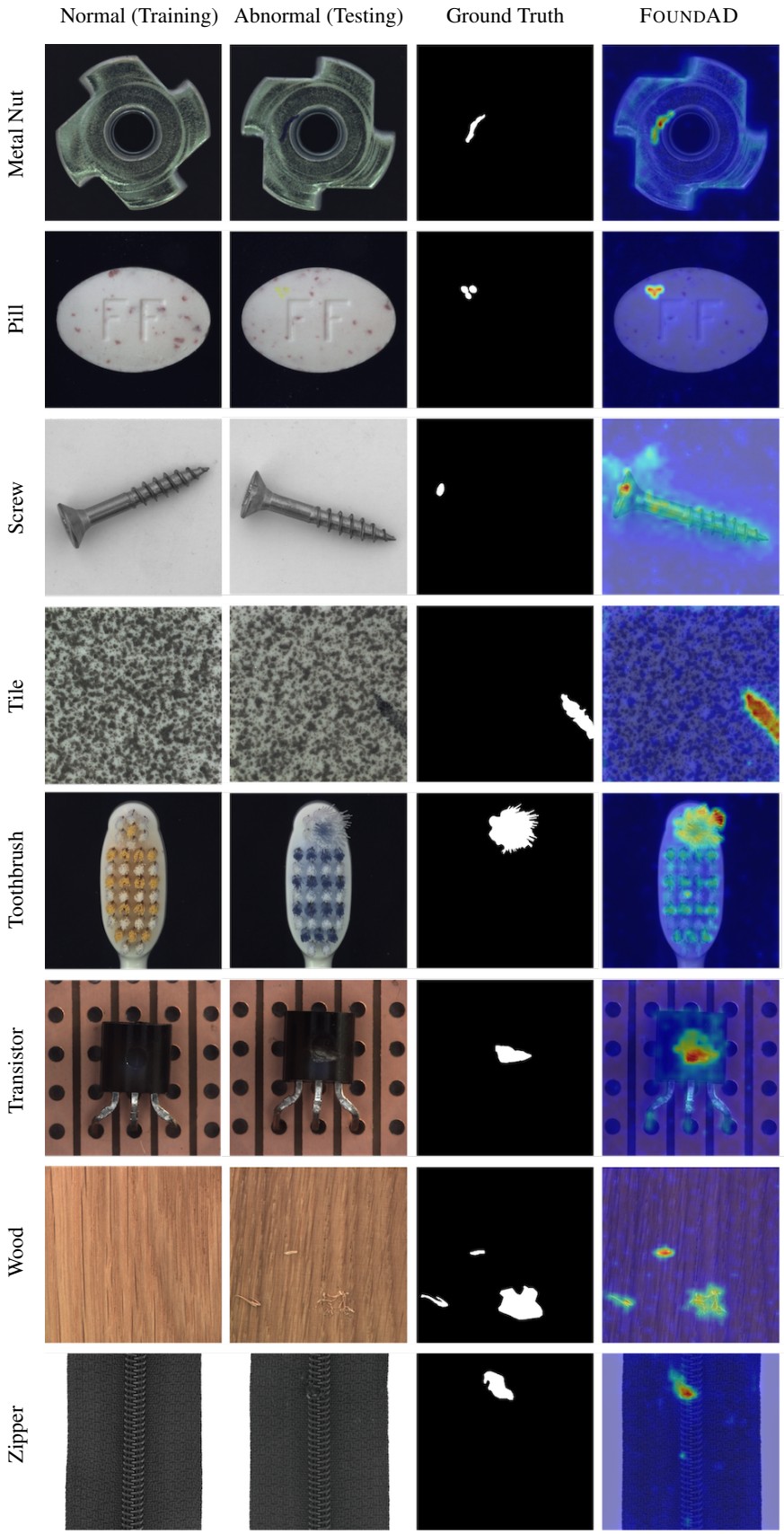

Figure 11: More Qualitative 1-shot Results on MVTec-AD (ii).

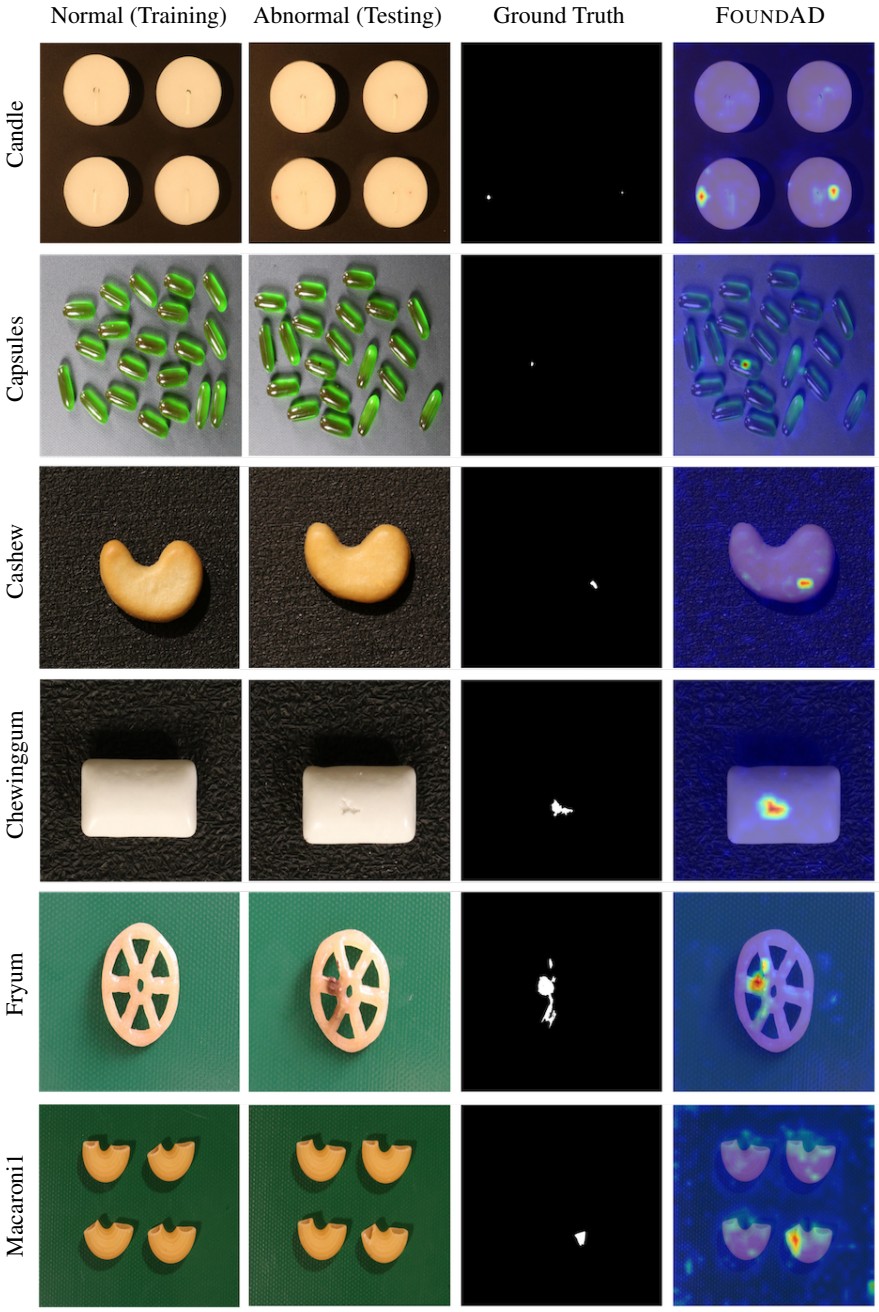

Figure 12: More Qualitative 1-shot Results on VisA (i).

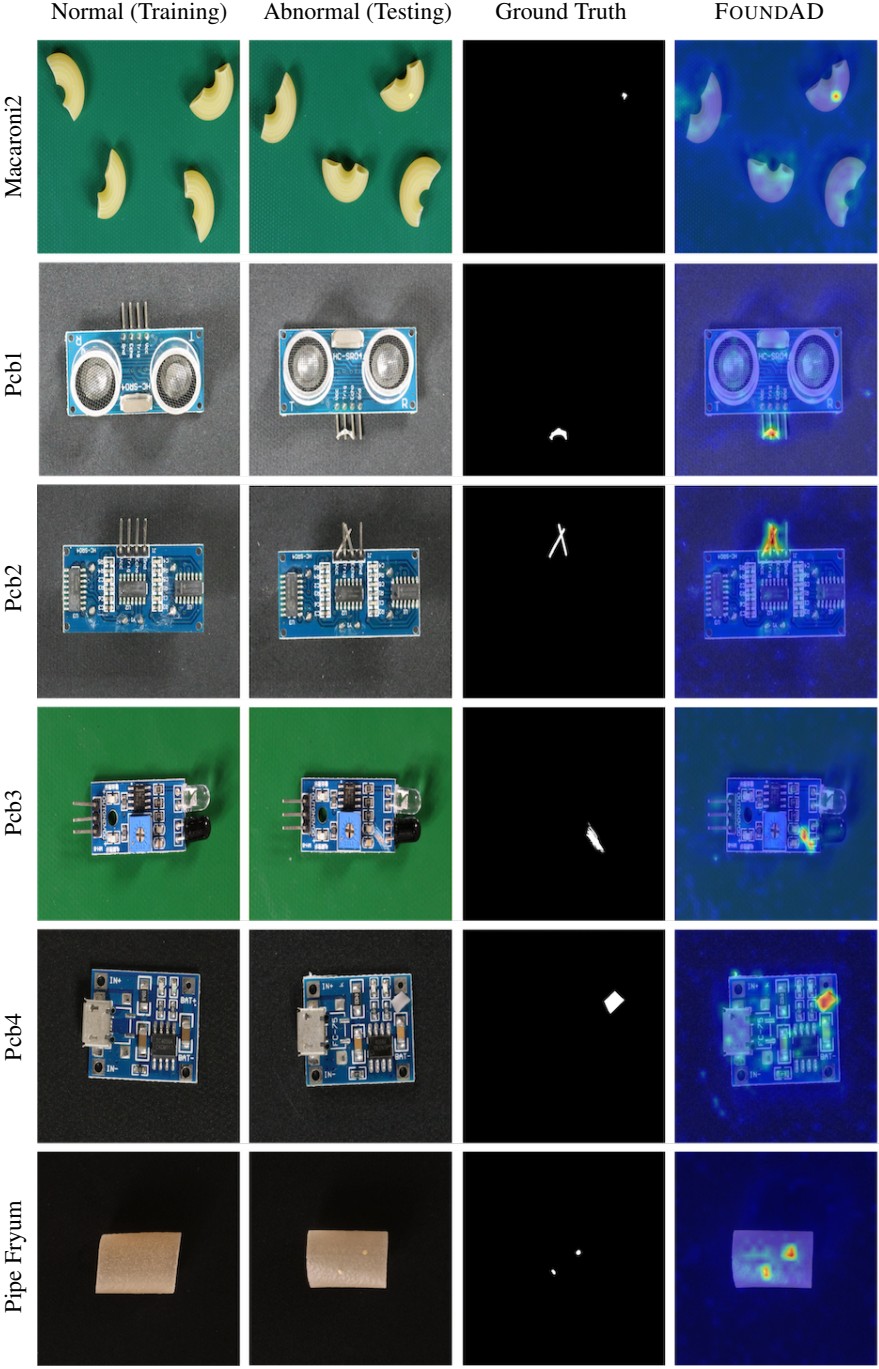

Figure 13: More Qualitative 1-shot Results on VisA (ii).

