# OpenReview forum: "Foundation Visual Encoders Are Secretly Few-Shot Anomaly Detectors"
_ICLR.cc/2026/Conference — ICLR 2026 Poster_

### Official Review · Reviewer_Bnw6 · 2025-10-25

**Soundness:** 3
**Presentation:** 4
**Contribution:** 3
**Rating:** 6
**Confidence:** 5

**Summary:**

This paper observes that in foundation visual encoders, the embedding distance naturally correlates with the amount of anomaly within an image, indicating that these pretrained models implicitly capture the structure of the normal image manifold. Building on this observation, the authors propose a few-shot anomaly detection framework that leverages this intrinsic property without retraining the encoder.

**Strengths:**

1. The paper is well written and easy to follow. The motivation, method design, and experimental setup are clearly articulated, making the overall contribution easy to understand.
2. The proposed method achieves impressive performance across multiple benchmarks, including the multi-class few-shot setting, the improvements over prior methods are consistent and substantial.
3. The proposed method introduces only a small projection network with minimal trainable parameters and fast inference. Also not relied on text prompts.

**Weaknesses:**

1. The method trains the projector using synthetic anomalies generated by CutPaste, which differ significantly from real-world defects in texture and scale. The paper does not provide a clear empirical justification for why this training signal generalizes well to real anomalies. The paper should include a more explicit discussion of this transferability.
2. Whether using more realistic synthetic data, e.g., anomalies generated by modern diffusion-based methods, would further improve or alter performance? An ablation in this direction would better explain the role of synthesis process in this framework.
3. The method uses features from a single mid-level layer of the foundation encoder. Have the authors considered aggregating multi-layer embeddings to capture both fine-grained and semantic anomalies?
4. The paper should include a baseline using the raw frozen encoder features without fine-tuning to see how much performance comes from the inherent foundation encoder.

**Questions:**

See above weaknesses. The method is conceptually simple and technically lightweight. Although the empirical results are strong, I would be more interested in a deeper discussion of the experimental setting and underlying factors.

---

> ### Author Response · Authors · 2025-11-19
> **Response to Reviewer Bnw6 (1/2)**
>
> **1. The reviewer would like to see more discussion on transferability from synthetic to real anomalies.** We agree that cut-and-paste anomalies differ in texture and scale from real ones, and we do not intend them to mimic every real anomaly type. In FoundAD, synthetic anomalies serve only as a *weak supervision* signal to teach the projector to “prefer” the natural image manifold encoded by the frozen foundation model.  The projector learns to separate on-manifold from off-manifold features in the latent space, rather than overfitting to the specific cut-and-paste pixel-level patterns. This transfers the anomaly detection problem into projecting off-manifold features back onto the normal manifold (cf. Figure 1). Empirically, Figure 2 provides evidence for this transferability by the PCA visualizations, where the synthetic anomaly features are displaced from the region of normal features in a very similar way to real anomalies, and the center plot shows a monotonic correlation between synthetic anomaly area and feature distance. All these evidences indicate that the synthetic corruptions in the latent space act as a meaningful surrogate signal for “off-manifold” behavior, even though their pixel appearance differs from real defects.
>
> **2. The reviewer suggests using diffusion-based synthesis.** We appreciate this idea and agree that more realistic synthetic data is an interesting direction to explore. In this work, however, one of our core motivations is to **maintain both high training and inference efficiency**, so we design the synthesis process to run on the fly during training with negligible overhead. A diffusion-based generator would substantially slow down training and require an additional large model, typically pre-trained on specific datasets, which conflicts with this design goal. We therefore choose cut-and-paste synthesis because it is simple, flexible, and can be easily applied to any image. Figure 2 also shows that the features of both cut-and-paste anomalies and real defects move away from normal features in latent space in a similar manner, while FoundAD already achieves strong results based on this point. These observations suggest that increasing pixel-level realism via heavy diffusion models is not the main bottleneck in the current framework.
>
> **3. The reviewer would like to see the performance based on multi-layer embeddings.** We take DINOv2 as an example. In the paper, our reported DINOv2 configuration uses the third-to-last layer (layer 10) in the 1-shot setting on MVTec-AD, achieving 95.2 / 97.4 / 96.4 / 92.5 on I-AUROC / AUPR / P-AUROC / PRO (cf. Table 3). In early experiments, we had also explored aggregating multi-layer embeddings in the same 1-shot setting: we concatenated and fused features from layers 3, 6, 9, and 12 of DINOv2 and trained the same projector. This significantly degraded performance, yielding 88.0 / 93.4 / 61.8 / 20.3 on I-AUROC / AUPR / P-AUROC / PRO. We suspect that mixing different abstraction levels makes it harder for a lightweight projector to learn a consistent normal manifold, while also increasing feature dimensionality and memory. These results support our design choice of using a single mid-level layer, which already provides a good balance between fine-grained detail and semantics. Nevertheless, we believe that extending FoundAD with more sophisticated multi-layer aggregation is an interesting direction for future work.

---

> ### Author Response · Authors · 2025-11-19
> **Response to Reviewer Bnw6 (2/2)**
>
> **4. The reviewer would like to see a baseline using raw foundation features without fine-tuning.** In FoundAD, the encoder is already kept **entirely frozen**; we do not apply any fine-tuning to it and only train a lightweight projector on top. Training-free baselines such as LogSAD in Table 2, which operate directly on frozen foundation encoders, already partially play this role.
>
> Meanwhile, we notice that AnomalyDINO (WACV’25) is a training-free, memory–bank–based method that operates directly on raw DINOv2 features in the `one-class-one-model` setting. We now additionally upgrade AnomalyDINO with DINOv3 and evaluate it on MVTec-AD and VisA. Please note that AnomalyDINO by default masks the primary object in the image from its background to reduce false-positive predictions, which we do not.
>
> **Table i. Comparison on MVTec-AD**
>
> | Shot  |Method| I-AUROC | AUPR | P-AUROC | PRO |
> |-------|--------|--------|-------|--------|------|
> | 1| AnomalyDINOv3 | 96.3    | 98.2   | 97.0    | 93.9  |
> | | FoundAD | 96.1    | 97.9   | 96.8    | 92.8  |
> | | FoundAD-single-class | 97.4| 98.3| 97.2| 93.6 |
> | 2| AnomalyDINOv3 | 96.9    | 98.3   | 97.1    | 94.2  |
> | | FoundAD | 96.9    | 98.3   | 97.0    | 93.2  |
> | | FoundAD-single-class | 98.1| 99.0| 97.5| 94.0 |
> | 4| AnomalyDINOv3 | 98.3    | 99.1   | 97.3    | 94.7  |
> | | FoundAD | 97.1   | 98.6   | 97.2    | 93.6  |
> | | FoundAD-single-class | 98.2| 99.1| 97.7| 94.3 |
>
> **Table ii. Comparison on VisA**
>
> | Shot  |Method| I-AUROC | AUPR | P-AUROC | PRO |
> |-------|--------|--------|-------|--------|------|
> | 1| AnomalyDINOv3 | 91.7    | 92.6   | 98.1    | 91.8  |
> | | FoundAD | 92.6    | 92.0   | 99.7    | 98.0  |
> | 2| AnomalyDINOv3 | 91.7    | 92.7   | 98.2    | 92.2  |
> | | FoundAD | 93.8    | 93.3 | 99.7	| 98.2  |
> | 4| AnomalyDINOv3 | 92.8    | 93.3   | 98.4    | 93.3  |
> | | FoundAD | 94.4	|94.0	|99.7	|98.4|
>
>
> The results show that, even though FoundAD is evaluated under the harder `multi-class-one-model` setting, it is just slightly inferior to AnomalyDINOv3 on MVTec-AD. When we switch FoundAD to `one-class-one-model`, it performs better than AnomalyDINOv3 with lower shots, and is on par with higher shots (see detailed classes in Tables vi-viii in Reviewer `BLKp`'s panel). *Moreover, FoundAD in `multi-class-one-model` surpasses AnomalyDINOv3 by a large margin on VisA across all metrics.* Overall, these experiments indicate that the learned projector provides good gains over using raw foundation features alone.

---

### Official Review · Reviewer_vtuY · 2025-11-01

**Soundness:** 3
**Presentation:** 3
**Contribution:** 3
**Rating:** 8
**Confidence:** 4

**Summary:**

FOUNDAD introduces a simple yet effective few-shot anomaly detection framework using frozen foundation encoders and a lightweight projector trained in latent space. The approach achieves strong results on standard industrial benchmarks with minimal supervision, outperforming more complex baselines. Its simplicity, efficiency, and robustness in multi-class settings make it attractive for practical deployment. While some limitations remain in generalization and synthetic anomaly reliance, the contribution is clear.

**Strengths:**

This paper proposes FOUNDAD, a simple yet effective framework for few-shot anomaly detection using frozen foundation encoders and a lightweight projector trained in latent space. The core idea is that pre-trained encoders already capture a natural image manifold, and deviations from it can indicate anomalies. This intuition is clearly demonstrated both conceptually and empirically.

The method is efficient,requiring significantly fewer parameters than recent baselines, yet achieves competitive or superior results across multiple benchmarks (MVTec-AD, VisA), especially in the low-shot regime. FOUNDAD also operates without text prompts or heavy multi-model setups, which makes it more practical for real-world industrial applications. Presentation is mostly clear, and the supplemental material is thorough, including failure cases and ablations.

**Weaknesses:**

The reliance on synthetically generated anomalies (via CutPaste-like methods) introduces a potential domain gap. While the authors argue that the projector generalizes well, the method struggles with unseen anomaly types like rotated objects or subtle defects. This raises concerns about robustness.
Additionally, the approach heavily depends on the choice of encoder—DINOv3 works well, but others perform worse. It remains unclear how well this method would transfer to domains less similar to natural images (e.g., medical or satellite imagery).
The few-shot setup could be explained more clearly. It’s not immediately obvious how support images are selected and used, especially in the multi-class setting.

**Questions:**

1. Have you considered stronger augmentation strategies or geometric alignment (e.g., from RegAD) to improve generalization to rotated or appearance-shifted anomalies?
2. Can you clarify the few-shot protocol? Are normal samples provided per class, or pooled globally?
3. Did you experiment with partially fine-tuning the encoder instead of freezing it entirely?
4. How sensitive is performance to the CutPaste threshold or other synthesis hyperparameters?

---

> ### Author Response · Authors · 2025-11-19
> **Response to Reviewer vtuY (1/2)**
>
> **1. The reviewer thinks there is a domain gap in cut-and-paste style synthesis.** We agree on this point. However, they serve only as a *weak signal* to learn a projector. Although the test anomalies differ substantially from the synthetic corruptions observed during training, they are similar to the synthesized ones in the latent space, as visualized by PCA in Figure 2. Furthermore, FoundAD still performs well across categories, suggesting that the projector does not overfit to the specific synthetic patterns but instead generalizes beyond them.
> We agree that certain types, such as arbitrarily rotated objects (e.g., screw), remain challenging. This limitation is explicitly discussed in the Supplementary Material. Addressing arbitrary orientations likely requires architectural changes such as rotation-equivariant or rotation-invariant modules (e.g., spatial transformer network), which we consider orthogonal to our main contribution about foundation encoders and feature projection. We view incorporating such pose-robust components on top of FoundAD as an interesting and promising direction for future work.
>
> **2. The reviewer would like to know whether the method is encoder-dependent.** Our framework treats an arbitrary foundation visual encoder as a pluggable backbone. As shown in Table 3, DINOv3 performs best; however, other strong dense encoders, such as DINOv2 and DINOSigLIP, also yield competitive results, while CLIP and WideResNet lag behind. This trend is consistent with prior work: encoders with richer representations naturally perform better at anomaly localization. Thus, the performance variation reflects the suitability of the backbone rather than the brittleness of FoundAD itself.
>
> **3. The reviewer asks about performance on domain-specific data such as medical images.** First, we would like to clarify the primary scope of this work. As stated in the Abstract (L1) and Methodology (L141–142), our motivation is to develop **few-shot anomaly detection for industrial or structural defects**. Our framework is built around a structural anomaly synthesis module and is evaluated on industrial benchmarks MVTec-AD and VisA. In the revision, we additionally include two more datasets, BTAD and DTD, where FoundAD consistently achieves competitive or superior performance under the same few-shot protocol.
>
> We agree with the reviewer that domain-specific data, such as medical images, is also worth exploring. To this end, we train the projector with DINOv3 on the Br35H brain MRI dataset, focusing on image classification, in the 1-/2-/4-shot settings, and compare against methods including the recent AdaptCLIP (AAAI’26). The results are as follows:
>
> **Table i. Comparisons on the Br35H dataset**
> | Shot  | Method    | I-AUROC | AUPR |
> |-------|-----------|---------------|--------------|
> | 1| WinCLIP   | 80.1          | 79.4         |
> |       | InCTRL    | 83.9          | 82.6         |
> |       | AdaptCLIP | 93.7          | 92.1         |
> |       | FoundAD   | 86.6          | 79.4         |
> | 2| WinCLIP   | 81.6          | 80.5         |
> |       | InCTRL    | 86.1          | 84.8         |
> |       | AdaptCLIP | 94.0          | 92.2         |
> |       | FoundAD   | 88.8          | 81.7         |
> | 4| WinCLIP   | 82.3          | 81.0         |
> |       | InCTRL    | 89.1          | 88.6         |
> |       | AdaptCLIP | 93.7          | 91.8         |
> |       | FoundAD   | 89.1          | 82.2         |
>
> As shown in the table, FoundAD achieves performance comparable to WinCLIP and InCTRL, but is inferior to AdaptCLIP. This suggests that, even though Br35H images deviate from the natural image manifold, the foundation encoder still provides a meaningful embedding on which our projector can operate. At the same time, as the reviewer points out, the remaining performance gap to AdaptCLIP indicates that the current foundation encoder is not fully adapted to the medical domain. In principle, plugging a domain-specific encoder (e.g., trained directly on medical images) into the same pipeline is straightforward and could further improve performance in such specialized settings. We will explicitly discuss this experiment and the limitations in the revised manuscript.
>
> **4. The reviewer would like to know how support images are selected.** In the $K$-shot setting, we randomly sample $K$ normal training images per category.  We pool all support images and train a single shared projector. For example, in a 1-shot setting on MVTec-AD, we select one image from each class (15 in total) and pool them together as our training data. As described in Sec. 3.2, we repeat this process under three random seeds and report the mean performance.

---

> ### Author Response · Authors · 2025-11-19
> **Response to Reviewer vtuY (2/2)**
>
> **5. The reviewer asks about the augmentations we used.** In early experiments, we tested various combinations of augmentation strategies when generating synthetic anomalies, including horizontal/vertical flipping, 90-degree rotations, color jittering, grayscale conversion, and Gaussian blurring. *The corresponding configurations can be found in the anonymous GitHub repository and in the supplementary zip files.* We did observe noticeable gains mainly on the most pose-sensitive classes (e.g., screw), while improvements on other classes were modest; all reported results in the paper already use these augmentations. This suggests that the current bottleneck lies less in the augmentation strategies and more in explicitly modeling pose/geometry. In line with the reviewer’s suggestion, we already note in Supplementary Material Section B (L720–730) that incorporating rotation-equivariant or alignment modules as in RegAD into FoundAD is a promising future direction, which we regard as an orthogonal extension to our core contribution of leveraging foundation visual encoders with feature projection.
>
> **6. The reviewer asks whether we performed partial fine-tuning of the encoder.** We did not. We intentionally keep the foundation encoder entirely frozen, as our goal is to show that off-the-shelf foundation visual encoders are already strong few-shot anomaly detectors once equipped with a lightweight projector.
>
> In early experiments, we tried fully fine-tuning the encoder on the same few normal examples and observed a substantial performance drop: I-AUROC decreased to 0.59 and P-AUROC to 0.55. This behavior is expected, as full fine-tuning on very few normal samples tends to overfit and distort the feature manifold learned during large-scale pre-training, which our method explicitly relies on. For this reason, we choose to freeze the encoder to preserve the natural image manifold and only learn a small projector on top. However, we believe that exploring carefully designed partial fine-tuning strategies is an interesting direction, but it is orthogonal to our core contribution.
>
> **7. The reviewer asks how the cut-and-paste threshold $\sigma$ affects FoundAD.** We originally set $\sigma = 0.5$ to roughly balance the ratio of normal vs. anomalous training samples. To verify robustness, we additionally tested $\sigma \in \{0.3, 0.4, 0.6, 0.7\}$ on MVTec-AD in the 1-shot setting (Seed 1), obtaining:
> | $\sigma$ | I-AUROC | AUPR  | P-AUROC | PRO  |
> |---------|:--------:|:------:|:--------:|:-----:|
> | 0.3      | 95.2    | 97.0  | 96.8    | 92.8 |
> | 0.4      | 94.9    | 97.0  | 96.8    | 92.7 |
> | 0.5      | 96.0    | 97.8  | 96.9    | 93.1 |
> | 0.6      | 95.6    | 97.5  | 96.9    | 92.9 |
> | 0.7      | 95.4    | 97.6  | 96.9    | 92.8 |
>
>
> I-AUROC and P-AUROC remain stable, with averages of about 95.4 and 96.9 and standard deviations below 0.4 and 0.05, respectively, and AUPR/PRO show similarly small variations. This indicates that performance is not sensitive to the choice of $\sigma$. We will include these results in the Supplementary Material.

---

### Official Review · Reviewer_BLKp · 2025-11-01

**Soundness:** 3
**Presentation:** 3
**Contribution:** 2
**Rating:** 4
**Confidence:** 5

**Summary:**

This paper proposes FOUNDAD, a lightweight few-shot anomaly detection method leveraging pre-trained foundation visual encoders like DINOv3 without text prompts. It introduces a nonlinear manifold projector that maps image embeddings onto a natural image manifold, enabling anomaly localization using only few normal samples. Experiments on MVTec-AD and VisA datasets show FOUNDAD achieves state-of-the-art performance with fewer parameters than competing methods.

**Strengths:**

- The proposed nonlinear manifold projector is simple yet effective, requiring minimal parameters and training, while achieving state-of-the-art few-shot anomaly detection performance.

- Evaluations across few-shot settings (1, 2, 4) demonstrate efficacy of the method.

**Weaknesses:**

- The idea of using latent feature distances for anomaly detection resembles prior projection or reconstruction-based methods. In particular, projecting anomaly feature back into normal manifold is a long established idea for anomaly detection, e.g. GAN or AutoEncoder based methods.

- The details of projector training is missing. For example, the number of training epochs, how to generate training data, etc. These are very important information because it remains elusive how the projector can be trained with a single image (1-shot). Moreover, the number of training epochs/iterations also reveal some key insights into the reason why the model is effective.

- The reliance on CutPaste-like synthesis might limit realism and generalization to complex industrial defects not represented by structural cut-paste anomalies. Although the performance on MVTec and VisA demonstrate the effectiveness, evaluation on more challenging datasets may better reveal the effectiveness.

- CutPaste augmentation is a very important component for this method. The original CutPaste work directly trained a classification model differentiating good from cut paste augmented pseudo defects. Therefore, a comparison with the original CutPaste method is essential to demonstrate the effectiveness. Importantly, the original CutPaste should enjoy the same foundation model as backbone.

- The proposed method benefits from the multi-class setting due to more data available for training the projector. It will further improve the significance if single-class setting is evaluated.

- Only two industrial datasets (MVTec-AD, VisA) are used, thus limiting the significance of conclusion.

**Questions:**

I encourage the authors to address the following issues in the rebuttal and I would adjust the rating based on the results.

- Please provide more details of training procedure and evaluate how the training procedure may affect the results.

- Additional datasets beyond MVTec and VisA are necessary to demonstrate the effectiveness.

- Please compare with original CutPaste with the same foundation model as backbone.

- More evaluations under single-class setting is necessary.

---

> ### Author Response · Authors · 2025-11-19
> **Response to Reviewer BLKp (1/3)**
>
> **1. The reviewer asks for clarification of the idea.** We agree that projecting anomaly features back into normal ones is a common practice in the community. However, our contribution differs in both task setting and mechanism.
> Previous reconstruction-based approaches, such as UniAD and SimpleNet, were developed for the full-shot setting and typically train end-to-end with anomaly jittering applied to all encoded features.
> This strategy relies on large training sets and tends to underperform in few-shot scenarios, where the model must learn to reconstruct normal features from stochastic jittered features with more training samples.
> In contrast, FoundAD is designed for the few-shot setting: we introduce anomaly features in random areas on top of foundation visual features, and highlight that the natural image manifold simplifies the task due to a large-scale pretraining of the encoder (cf. L73-75).
>
> **2. The reviewer asks for details on training epochs and training data.** We set a constant learning rate of 0.001 and trained for 2000 epochs in the 1-shot setting and 2500 epochs in the 2- and 4-shot settings, determined by monitoring the trend of the training loss. Each image randomly chooses a data augmentation, including horizontal/vertical flipping, 90-degree rotations, color jittering, grayscale conversion, and Gaussian blurring. The anomaly synthesis happens on the fly during training for more efficient data processing and augmentation. The foundation features enjoy a natural image manifold and possess strong cross-instance semantic and geometric understanding per category (cf. L83-84), which enables effective 1-shot demonstration for other test samples.
>
> **3. The reviewer asks how the training procedure affects the results.** We illustrate this with the data augmentation study.
>
> **Table i. Ablation on augmentation strategies.**
> | Augmentation                            | I-AUROC  | AUPR    | P-AUROC | PRO    |
> |--------------------------------|----------|---------|---------|--------|
> | hflip_vflip                    | 95.8   | 97.9  | 97.0  | 92.9 |
> | hflip_vflip_rotate             | 96.1   | 97.8   | 96.9   | 93.0  |
> | hflip_vflip_rotate_cj_gray     | 96.1    | 97.9   | 96.9   | 93.0  |
> | hflip_vflip_rotate_cj_gray_blur| 96.1    | 97.9   | 97.0   | 93.1  |
>
> We observe that rotation is the most critical augmentation: going from hflip_vflip to hflip_vflip_rotate noticeably improves all metrics. Adding further photometric transforms (color jitter, gray, blur) on top of rotation brings only incremental gains.

---

> ### Author Response · Authors · 2025-11-19
> **Response to Reviewer BLKp (2/3)**
>
> **4. The reviewer would like to see the performance of more datasets.** We conduct experiments on the BTAD and DTD datasets, using DINOv3 as the backbone. BTAD focuses on body and surface defects of industrial products, while DTD specializes in textures. We also compare our results with those of a newly accepted baseline, AdaptCLIP (AAAI 2026). The results are as follows:
>
> **Table ii. Results on BTAD**
> | Shot   | Method      | I-AUROC | AUPR | P-AUROC | PRO |
> |--------|-------------|--------|-------|--------|------|
> | 1| WinCLIP     | 84.4    | 80.5   | 95.6    | 66.6  |
> |        | InCTRL      | 88.5    | 83.4   | 96.6    | 71.0  |
> |        | AnomalyCLIP | 88.5    | 74.2   | 95.3    | 75.5  |
> |        | AdaptCLIP   | 93.4    | 95.8   | 96.6    | 77.3     |
> |        | FoundAD     | 96.4    | 96.5   | 98.1    | 84.1  |
> | 2| WinCLIP     | 85.8    | 82.5   | 95.7    | 66.8  |
> |        | InCTRL      | 86.2    | 81.6   | 96.7    | 71.5  |
> |        | AnomalyCLIP | 89.2    | 75.4   | 95.5    | 76.0  |
> |        | AdaptCLIP   | 93.4    | 95.9   | 96.7    | 77.8     |
> |        | FoundAD     | 97.3    | 96.8   | 98.2    | 83.3  |
> | 4| WinCLIP     | 87.8    | 88.1   | 95.9    | 67.0  |
> |        | InCTRL      | 67.5    | 80.9   | 96.8    | 71.4  |
> |        | AnomalyCLIP | 90.5    | 77.5   | 95.7    | 76.6  |
> |        | AdaptCLIP   | 93.3    | 96.4   | 96.8    | 78.2    |
> |        | FoundAD     | 95.5    | 96.7   | 98.3    | 82.7  |
>
> **Table iii. Results on DTD**
> | Shot   | Method      | I-AUROC | AUPR | P-AUROC | PRO |
> |--------|-------------|--------|-------|--------|------|
> | 1| WinCLIP     | 97.9    | 99.0   | 96.5    | 89.7  |
> |        | InCTRL      | 97.9    | 98.9   | 98.6    | 95.6  |
> |        | AnomalyCLIP | 98.0    | 99.2   | 97.6    | 93.5  |
> |        | AdaptCLIP   | 98.0    | 99.1   | 97.4    | 92.8    |
> |        | FoundAD     | 99.2    | 99.7   | 98.9    | 97.5  |
> | 2 | WinCLIP     | 98.1    | 99.1   | 96.6    | 89.8  |
> |        | InCTRL      | 98.3    | 99.1   | 98.7    | 95.6  |
> |        | AnomalyCLIP | 98.2    | 99.3   | 97.7    | 93.7  |
> |        | AdaptCLIP   | 97.4    | 99.2   | 97.6    | 92.9   |
> |        | FoundAD     | 99.2    | 99.7   | 98.9    | 97.5  |
> | 4| WinCLIP     | 98.2    | 99.2   | 96.8    | 90.4  |
> |        | InCTRL      | 97.7    | 98.3   | 98.7    | 95.9  |
> |        | AnomalyCLIP | 98.4    | 99.4   | 97.8    | 93.8  |
> |        | AdaptCLIP   | 98.5    | 99.3   | 97.8    | 93.2   |
> |        | FoundAD     | 99.3    | 99.7   | 99.0    | 97.6  |
>
>
> We can see that FoundAD surpasses all baselines on both datasets, indicating the advantage of learning in the latent space within our architecture. We will include these results in the manuscript.
>
> Meanwhile, we also provide the comparison with AdaptCLIP on MVTec-AD and VisA as follows:
>
> **Table iv. Compare to AdaptCLIP on MVTec**
> | Shot   | Method      | I-AUROC | AUPR | P-AUROC | PRO |
> |--------|-------------|--------|-------|--------|------|
> | 1| AdaptCLIP     | 94.5 | 97.5 | 94.3 | 89.5 |
> | | FoundAD     | 96.1| 97.9 |96.8| 92.8|
> | 2| AdaptCLIP     | 95.7 | 97.9 | 94.5 | 90.3 |
> | | FoundAD     | 96.8| 98.3| 97.0 |93.3|
> | 4| AdaptCLIP     | 96.6 | 98.4 | 94.8 | 90.9 |
> | | FoundAD     | 97.1| 98.6| 97.2| 93.5|
>
> **Table v. Compare to AdaptCLIP on VisA**
> | Shot   | Method      | I-AUROC | AUPR | P-AUROC | PRO |
> |--------|-------------|--------|-------|--------|------|
> | 1| AdaptCLIP     | 90.5 | 92.3 | 96.8 | 90.7 |
> | | FoundAD     | 92.6| 92.0| 99.7| 98.0|
> | 2| AdaptCLIP     | 92.2 |93.6 | 97.1|91.2 |
> | | FoundAD     | 93.5| 93.0 |99.7 |98.0 |
> | 4| AdaptCLIP     | 93.1| 94.3| 97.3| 91.7 |
> | | FoundAD     | 94.4 | 94.0 |99.7 |98.4 |
>
> **5. The reviewer asks for an experiment for the original CutPaste with DINOv3.** We would like to clarify two points:
> > (1) In our framework, **we do not adopt the original CutPaste pipeline**, and we only use the cut-and-paste operation as a simple synthesis mechanism on a few normal support images. These synthetic anomalies are not used as a separate class for classification; they serve only as a weak off-manifold supervision signal to train a small projector in latent space that pulls normal features toward the normal manifold and pushes anomaly ones away.
>
> > (2) CutPaste is a full-shot method. Reproducing the original CutPaste pipeline with a foundation encoder in our few-shot setting would effectively amount to designing a new variant rather than the original method, and **is orthogonal to our main contribution**. Instead, we focus on comparisons with strong modern baselines (LogSAD, IIPAD, now AdaptCLIP, etc.) under the few-shot protocol.
>
> To better address the concern, could the reviewer please clarify how you would ideally like us to compare with CutPaste in more detail (e.g., a particular variant or evaluation setting)?

---

> > ### Comment · Reviewer_BLKp · 2025-11-27
> >
> > Thanks for the clarification on CutPaste, which prompted me to take a closer look at the method. CutPaste learns a representation by distinguishing cut-paste augmented images from normal ones, and relies on GDE for anomaly scoring at test time. I agree that it is not directly comparable in this context.
> >
> > However, this recap raises an important question. FoundAD assumes that a **frozen DINO-V3 encoder** can already capture meaningful differences between normal and abnormal regions in its feature maps, otherwise the method would not work. If these differences are indeed well-captured, **why do density-based methods, such as PatchCore or parametric models like GDE, perform so much worse**, as reported in Table 2 of the manuscript? Could this gap be due to not using the same encoder (DINO-V3) or to insufficient data augmentation when constructing the memory bank for PatchCore?
> >
> > More broadly, **do the competing baselines (not only PatchCore) suffer simply because they are built upon weaker visual encoders compared to FoundAD?**

---

> ### Author Response · Authors · 2025-11-19
> **Response to Reviewer BLKp (3/3)**
>
> **6. The reviewer believes that the one-class-one-model performance could deteriorate.** We would like to clarify that the `multi-class-one-model` setting is in fact more challenging than training a separate model per class. As shown in our experiments, several baselines suffer noticeable performance drops when moving from `one-class-one-model` to `multi-class-one-model`: WinCLIP’s I-AUROC decreases by around 1.3% on average, PromptAD by around 5.7%, SPADE by around 14.3%, etc. This is because a single network must learn to detect diverse anomaly types across many categories from only a few samples per class, rather than specializing in a single category. We train projectors on MVTec-AD for 15 classes in the 1-/2-/4-shot setting (45 models in total), and provide results here:
>
> **Table vi. Single-class results on MVTec-AD in the 1-shot setting**
> | Class       | I-AUROC | AUPR | P-AUROC | PRO |
> |------------|---------|--------|---------|-------|
> | bottle     | 100.0   | 100.0  | 98.5    | 96.4  |
> | cable      | 92.3    | 96.1   | 95.0    | 89.7  |
> | capsule    | 96.3    | 99.2   | 98.5    | 94.8  |
> | carpet     | 100.0   | 100.0  | 99.5    | 98.6  |
> | grid       | 100.0   | 100.0  | 99.5    | 97.3  |
> | hazelnut   | 100.0   | 100.0  | 99.6    | 95.6  |
> | leather    | 100.0   | 100.0  | 99.4    | 99.1  |
> | metal_nut  | 100.0   | 100.0  | 93.2    | 92.6  |
> | pill       | 98.2    | 99.7   | 94.7    | 97.1  |
> | screw      | 89.3    | 95.3   | 98.8    | 94.0  |
> | tile       | 100.0   | 100.0  | 97.2    | 95.5  |
> | toothbrush | 100.0   | 100.0  | 99.3    | 94.2  |
> | transistor | 87.5    | 84.8   | 88.3    | 65.3  |
> | wood       | 98.7    | 99.6   | 97.0    | 96.6  |
> | zipper     | 99.2    | 99.8   | 98.9    | 96.6  |
> | **mean**   | **97.4**| **98.3**| **97.2**| **93.6** |
>
> **Table vii. Single-class results on MVTec-AD in the 2-shot setting**
>
> | Class       | I-AUROC | AUPR | P-AUROC | PRO |
> |------------|---------|--------|---------|-------|
> | bottle     | 99.6    | 99.9   | 98.5    | 96.6  |
> | cable      | 92.8    | 96.7   | 95.2    | 90.0  |
> | capsule    | 97.8    | 99.6   | 98.8    | 96.4  |
> | carpet     | 100.0   | 100.0  | 99.5    | 98.9  |
> | grid       | 100.0   | 100.0  | 99.5    | 97.4  |
> | hazelnut   | 100.0   | 100.0  | 99.6    | 95.1  |
> | leather    | 100.0   | 100.0  | 99.4    | 98.9  |
> | metal_nut  | 100.0   | 100.0  | 95.8    | 95.0  |
> | pill       | 98.5    | 99.7   | 95.5    | 97.4  |
> | screw      | 89.5    | 95.2   | 98.8    | 94.0  |
> | tile       | 100.0   | 100.0  | 97.3    | 94.8  |
> | toothbrush | 100.0   | 100.0  | 99.3    | 94.5  |
> | transistor | 95.3    | 94.4   | 88.9    | 67.5  |
> | wood       | 98.9    | 99.7   | 96.8    | 96.7  |
> | zipper     | 99.5    | 99.9   | 98.9    | 96.1  |
> | **mean**   | **98.1**| **99.0**| **97.5**| **94.0** |
>
> **Table viii. Single-class results on MVTec-AD in the 4-shot setting**
>
> | Class       | I-AUROC | AUPR | P-AUROC | PRO |
> |------------|---------|--------|---------|-------|
> | bottle     | 99.8    | 100.0  | 98.7    | 96.5  |
> | cable      | 92.9    | 96.6   | 95.7    | 90.9  |
> | capsule    | 97.3    | 99.4   | 98.8    | 95.5  |
> | carpet     | 100.0   | 100.0  | 99.5    | 98.3  |
> | grid       | 100.0   | 100.0  | 99.5    | 97.5  |
> | hazelnut   | 100.0   | 100.0  | 99.6    | 94.3  |
> | leather    | 100.0   | 100.0  | 99.4    | 99.0  |
> | metal_nut  | 100.0   | 100.0  | 97.1    | 96.0  |
> | pill       | 99.2    | 99.8   | 96.3    | 97.8  |
> | screw      | 88.6    | 94.8   | 98.8    | 93.3  |
> | tile       | 100.0   | 100.0  | 97.3    | 95.3  |
> | toothbrush | 100.0   | 100.0  | 99.4    | 96.5  |
> | transistor | 96.4    | 95.7   | 89.7    | 70.2  |
> | wood       | 98.9    | 99.6   | 96.7    | 96.5  |
> | zipper     | 99.7    | 99.9   | 99.1    | 97.4  |
> | **mean**   | **98.2**| **99.1**| **97.7**| **94.3** |
>
> Compared to the original Table 7 in the Supplementary Material, we can see that all mean results are better.

---

> ### Author Response · Authors · 2025-11-27
> **Discussion I**
>
> We thank the reviewer for their kind reply!
>
> The methods in Table 2 were derived from IIPAD, as written in the caption. Yes, we believe the reason why methods, such as PatchCore and FastRecon, suffer from poor performance is likely because the pretrained WideResNet is weaker than the DINO series, while other CLIP-based methods using text assistance cannot operate with only DINO features. We also observed a similar trend when we tested WideResNet in FoundAD. As shown in Table 3 of the ablation study, the 1-shot results from WideResNet are also not satisfactory. We analyzed the reasons in L383-384: *"WideResNet shows the weakest performance, indicating that traditional CNNs, even pretrained, struggle to generalize well under minimal supervision."*
>
> Actually, this is also the concern from Reviewer `Bnw6`, who would like to see a baseline using raw DINO features without any training. We believe that Reviewer `Bnw6` would like us to design a pipeline similar to PatchCore, but utilizing raw DINO features. Luckily, we found a baseline AnomalyDINO, which uses DINOv2 as the feature encoder, and operates in the same pattern. We upgrade its performance with DINOv3 and compare it to FoundAD under 1,2,4-shots (see detailed numbers of Table i-ii in Reviewer `Bnw6`'s panel 2/2).
>
> On MVTec-AD, FoundAD outperforms AnomalyDINOv3 both in the `one-class-one-model` setting, particularly with fewer shots, and is on par with it when using more shots. Moreover, FoundAD surpasses AnomalyDINOv3 by a large margin on VisA across all metrics, even though FoundAD is in the `multi-class-one-model` setting and AnomalyDINOv3 is still in the  `one-class-one-model` setting (we clarified why `multi-class-one-model` is more difficult in response 3/3). The results indicate that training a manifold projector is necessary for achieving higher results. For more information, please check our response to Reviewer `Bnw6` (2/2).
>
> Please feel free to let us know if any future issues arise (e.g., clarifications, experiments, writings), so that we can engage in the discussion in time and strengthen our manuscript.

---

> > ### Comment · Reviewer_BLKp · 2025-11-28
> >
> > Thanks for the additional evaluations. The results indeed suggest stronger backbone is a very important factor contributing to the performance. Therefore, I recommend the authors to **explicitly state the visual encoder for each method in the comparisons.** This is very important for future reseachers to follow up and understand the key bottleneck. Moreover, AnomalyDINO is a typical memory bank based method, thus by default it is nontrivial for AnomalyDINO to exploit either synthesized (e.g. CutPaste) or real anomalies. To my understanding, the advantages of FoundAD compared with memory bank based method are two-folds.
> >
> > i) The projector learns to project outlier (CutPaste synthesized) features back to inlier (normal image) features. Thus the **projector is implicitly learning the distribution of normal samples**.
> >
> > ii) The **projector is able to exploit synthesized anomalies** and learn the difference between synthesized anomalies and normal ones.
> >
> > The observations suggest the authors may consider an additional evaluation to verify the two hypotheses.
> >
> > i) The advantage of projector is coming from knowing the distribution of normal samples.
> >
> > ii) The advantage is coming from knowing the distribution of potential anomalies (e.g. synthesized defects) and how it deviates from distribution of normal samples (this is the original argument of this paper).
> >
> > I would encourage the authors to try alternative data augmentations or simply using real anomalies to swap the CutPaste defects (maybe the closest normal sample can be used as the reference normal sample for training in this case). The results can well explain the above two hypotheses.

---

> ### Author Response · Authors · 2025-11-28
> **Discussion II (1/2) - Thanks for acknowledging the advantages**
>
> We are glad that the reviewer acknowledges that FoundAD has two advantages over the memory bank-based methods. Indeed, the projector is able to implicitly learn the distribution of normal samples.
>
> However, instead of learning the difference between synthesized anomalies and normal samples, the projector is **learning the difference between the latent spaces of anomalies and normality**, regardless of whether the anomalies are synthesized or real. Our Figure 2 qualitatively supports this via a PCA visualization: the embeddings of real and synthesized anomalies reside in a similar feature space, and they are differentiated from the natural image manifold where normality lies (we made a teaser in Figure 1). This suggests that the foundation visual encoder treats synthesized and real defects in a comparable way, enabling the projector to learn the latent-space separation between synthetic anomalies and normality and then generalize this separation to real anomaly detection.

---

> ### Author Response · Authors · 2025-11-28
> **Discussion II (2/2) - Alternative synthesis**
>
> The reviewer would like to see whether the alternative anomaly synthesis could also work with FoundAD and suggest using real anomalies. We need to first clarify that our goal is **few-shot anomaly detection**, which belongs to unsupervised learning, and real anomalies are not accessible.
>
> We propose an alternative solution: instead of using cut-and-paste synthesis in pixel space, we synthesize anomalies directly in feature space via *Feature Dropout*. This acts as a natural anomaly feature generator: by randomly discarding subsets of feature dimensions and mixing them with the remaining ones, the resulting representations deviate from the normal feature manifold, providing off-manifold examples for training the projector. Specifically, we first prepare two identical normal samples $I$. We then encode normal samples by DINOv3 to obtain $\theta_a(I)$ and $\theta_b(I)$, where $\theta_a(I)=\theta_b(I)$. We employ Dropout with a ratio of $0.2$ on $\theta_a(I)$ to have $\operatorname{Syn}(\theta_a(I))$ and send them to the projector to obtain $\phi(\operatorname{Syn}(\theta_a(I)))$. We minimize the loss $D(\phi(\operatorname{Syn}(\theta_a(I))), \theta_b(I))$.
>
> We evaluate this variant in the 1-shot setting on MVTec-AD and VisA (same seed as the one in Supplementary Material Section C Table 5):
>
> **Table i. 1-shot results on MVTec-AD**
> | Synthesis       | I-AUROC | AUPR | P-AUROC | PRO |
> |------------|---------|--------|---------|-------|
> |cut-and-paste (in the paper) | 96.0 | 97.8 | 96.9 | 93.1|
> |Dropout | 95.7|97.3|96.8|92.8|
>
> **Table ii. 1-shot results on VisA**
> | Synthesis       | I-AUROC | AUPR | P-AUROC | PRO |
> |------------|---------|--------|---------|-------|
> |cut-and-paste (in the paper) | 93.4 | 92.7 | 99.7 | 98.1|
> |Dropout | 92.8 | 92.3 | 99.7 | 98.3|
>
> As shown in Tables i and ii, the results with *Feature Dropout* are very close to those with the original cut-and-paste synthesis. This indicates that FoundAD is robust to the specific choice of synthesis mechanism and can maintain performance as long as the projector is trained on off-manifold features, further emphasizing our contribution of learning the latent-space separation between anomalies and normality. This is also acknowledged by the reviewer.

---

### Author Response · Authors · 2025-11-19
**Acknowledgements and Clarifications**

We sincerely thank all reviewers for their thoughtful and constructive feedback! In particular, we appreciate that Reviewers `BLKp` and `vtuY` found FoundAD to be **“simple yet effective”**, and that Reviewer `Bnw6` considered the “motivation, method design, and experimental setup” to be **“clearly articulated”**. Before addressing the individual comments, we first recap our main contributions:

**Recap:**
 > (1) We empirically demonstrate a strong correlation between embedding distance and anomaly amount for frozen foundation visual encoders.

 > (2) We propose a feature projection that exploits the property in point (1) to separate normal from anomalous features in the latent space.

 > (3) We achieve competitive few-shot anomaly detection performance without any textual supervision.

 > (4) Our method is lightweight and attains higher efficiency than previous state-of-the-art methods.

We appreciate the reviewers’ attention to detailed quantitative comparisons and have added several experiments accordingly. We hope that our additional experiments will be helpful during the discussion phase. At the same time, we would like to emphasize that the key contribution lies in the *underlying observations* and the *framework*,  which demonstrate that foundation visual encoders, with a small projector trained on simple synthetic anomalies, already constitute a strong and efficient few-shot anomaly detector. We hope that this **conceptual perspective**, together with the **competitive results** across multiple datasets, will be taken into account in the final evaluation.

---

### Author Response · Authors · 2025-11-24
**Follow-up on our responses**

Dear reviewers,

Thank you again for reviewing our paper. We are encouraged by the positive feedback and have carefully addressed the concerns raised as much as we could. We are now preparing the revised version.

We understand that everyone is busy, but we would greatly appreciate it if you could take a moment to look over our responses and let us know of any remaining issues. We would be more than happy to engage in further discussion with you!

Best regards,
The Authors

---

### Author Response · Authors · 2025-11-29
**Summary**

Dear AC and the new AC,

We feel sorry about this information leak issue, and we hope the venues can recover from it. We would like to thank the original AC for taking care of our submission, and we really appreciate the new AC for taking additional time to recheck our submission. We now summarize the review process so far.

In the original reviews:

> Reviewer `BLKp` would like to ask for: 1. clarifications of the idea; 2. training details; 3. the effect brought by the training procedure; 4. the performance on more datasets; 5. clarifications of CutPaste; and 6. one-class-one-model performance.

> Reviewer `vtuY` would like to know: 1. whether there is a domain gap brought by cut-and-paste style synthesis; 2. whether the method is encoder-dependent; 3. the performance on domain-specific data; 4. how support images are selected; 5. data augmentation we used; 6. whether we performed partial fine-tuning of the encoder; and 7. the effect brought by the cut-and-paste threshold.

> Reviewer `Bnw6` would like to suggest: 1. more discussion on transferability from synthetic to real anomalies; 2. using diffusion-based synthesis; 3. a model based on multi-layer embeddings; and 4. a baseline using raw foundation features without finetuning.

We believe that we have addressed all these concerns. Based on our responses, Reviewer `BLKp`  participated in the discussion.

> In the first round, Reviewer `BLKp` has a new question about whether methods based on, for example, memory banks suffer because of a weaker encoder. We answer the question by referring to the manuscript and cross-referencing the experiments in Reviewer `Bnw6` panel. Reviewer `BLKp` is happy with the answer and suggests that we emphasize these comments in the updated manuscript.

> In the second round, Reviewer `BLKp` acknowledges the advantages of our method and brings another question about alternative anomaly synthesis. We conduct a simple experiment that replaces cut-and-paste synthesis with feature dropout. We believe that the answer addresses the question.

Overall, we believe that the reviewers are positive about our work. We hope the new AC is able to check each reviewer's panel for more details for the final decision.

Best,
Your Authors

---

### Meta-Review · Area_Chair_NYA6 · 2026-01-03

**Summary:**

The paper proposes an anomaly detection framework utilizing foundation model features and a synthetic anomaly generation strategy (Cut-and-Paste). I recommend Accepting this paper. The authors have provided a robust rebuttal that effectively addressed the reviewers' concerns regarding the training details, encoder dependency, and the validity of the synthesis strategy.

**Reviewer Concerns:**

The authors should ensure that the additional discussions generated during the rebuttal，specifically the comparison with feature dropout synthesis and the analysis of encoder dependency，are incorporated into the final manuscript as promised.

**Reviewer Scores:**

The reviewers'attitude is positive, particularly following the active discussion.

---

### Decision · Program_Chairs · 2026-01-26

Accept (Poster)